# Using the IBM SPSS SW Tool with Wavelet Transformation for CO_2_ Prediction within IoT in Smart Home Care

**DOI:** 10.3390/s19061407

**Published:** 2019-03-21

**Authors:** Jan Vanus, Jan Kubicek, Ojan M. Gorjani, Jiri Koziorek

**Affiliations:** Department of Cybernetics and Biomedical Engineering, Faculty of Electrical Engineering and Computer Science, VSB—Technical University of Ostrava, Ostrava 70800, Czech Republic; jan.kubicek@vsb.cz; (J.K.) ojan.majidzadeh.gorjani@vsb.cz (O.M.G.); jiri.koziorek@vsb.cz (J.K.)

**Keywords:** Smart Home Care (SHC), monitoring, prediction, trend detection, Artificial Neural Network (ANN), Radial Basis Function (RBF), Wavelet Transformation (WT), SPSS (Statistical Package for the Social Sciences) IBM, IoT (Internet of Things), Activities of Daily Living (ADL)

## Abstract

Standard solutions for handling a large amount of measured data obtained from intelligent buildings are currently available as software tools in IoT platforms. These solutions optimize the operational and technical functions managing the quality of the indoor environment and factor in the real needs of residents. The paper examines the possibilities of increasing the accuracy of CO_2_ predictions in Smart Home Care (SHC) using the IBM SPSS software tools in the IoT to determine the occupancy times of a monitored SHC room. The processed data were compared at daily, weekly and monthly intervals for the spring and autumn periods. The Radial Basis Function (RBF) method was applied to predict CO_2_ levels from the measured indoor and outdoor temperatures and relative humidity. The most accurately predicted results were obtained from data processed at a daily interval. To increase the accuracy of CO_2_ predictions, a wavelet transform was applied to remove additive noise from the predicted signal. The prediction accuracy achieved in the selected experiments was greater than 95%.

## 1. Introduction

An intelligent building is one that is responsive to the requirements of occupants, organizations, and society. An intelligent building requires real-time information about its occupants so that it can continually adapt and respond [1]. Intelligent buildings respond to the needs of occupants and society, promoting the well-being of those living and working in them [2]. The researchers point out that the Adaptive House (the concept of a home which programs itself), Learning Homes and Attentive Homes must be programmed for a particular family and home and updated in line with changes in their lifestyle. The system monitors actions taken by the residents and looks for patterns in the environment which reliably predict these actions, where a neural network learns these patterns and the system then performs the learned actions automatically for improving the Quality of Life (QoL) [3]. Privacy, reliability and false alarms are the main challenges to be considered for the development of efficient systems to detect and classify the Activities of Daily Living (ADL) and Falls [4]. In order to provide a user-friendly environment for the management of the operational and technical functions along with providing support for the independent housing of senior citizens and disabled persons in buildings indicated as Smart House Care (SHC), it is necessary to make appropriate visualization of the technological process as required by the users with the possibility of the indirect monitoring of the seniors’ life activities based on the information obtained from the sensors used for the common management of the operational and technical functions in SHC. The properly devised visualization complements the final correct functionality of the intelligent building. Beaudin et al. described the use of computational and sensor technology in intelligent buildings with a focus on health monitoring in the households. This work comprises of a visualization for displaying health data and a proposal for improving the health and wellbeing of the users [5]. Booysen et al. explored machine-machine communication (M2M) to address the need for autonomous control of remote and distributed mobile systems in intelligent buildings [6]. Basu et al. used miniature wireless sensors in the wireless network to track and recognize the behavior of persons in the house. The visualization includes sensor data from the building to capture the duration of the activities [7]. Fleck et al. described a system based on intelligent cameras for 24-h monitoring and supervision of senior citizens. On this occasion, visualization is used to display relevant life information in this intelligent environment, which includes the evaluation of the seniors’ position and recognition of life activities [8]. The current trend for processing large volumes of measured quantities using Soft Computing Methods (SC) [9,10] is to use the available Big Data Analysis tools within the IoT platform [11,12].

The Internet of Things, shortly known as IoT, can be assumed as an integration layer which creates an interconnection of several physical devices, sensors, actuators, and controllers [13]. In simple words, the IoT allows objects other than computers or smartphones to use the Internet for sending and receiving information [14]. The number of connected IoT devices is rapidly growing. This growth could be the result of a very wide range of applications, ranging from basic home appliances and security systems to more sophisticated applications. For example, Xu et al. used IoT in order to construct a real-time system monitoring system for micro-environment parameters such as temperature, humidity, PM10 and PM2.5 [15]. Q. Min et al. suggested using IoT for monitoring of discrete manufacturing process based on IoT [16]. Wang et al. presented a feasible and reliable plantation monitoring system based on the Internet of Things, that combined the wireless sensor network, embedded development, GPRS communication technology, web service, and Android mobile platform [17]. Windarto et al. presented an application of IoT by implementing automation of lights and door in a room [18]. Coelho et al. used IoT to collect data from multiple heterogeneous sensors that were providing different types of information at a variety of locations in a smart home [19]. Data collection in this kind of examples usually results in big data. The Oxford dictionary defines big data as “extremely large data sets that may be analyzed computationally to reveal patterns, trends, and associations, especially relating to human behavior and interactions” [20]. One of the possibilities that come with big data processing is a predictive analysis which can provide predictions about the future or otherwise unknown events. Predictive analysis offers a wide range of applications such as social networking, healthcare, mobility, insurance, finance marketing, etc. Nyce suggested to use the predictive analysis for risk identifications and probabilities in order to provide an appropriate insurance rate, his method took advantage of marketing records, underwriting records, and claims records [21]. Predictive analysis includes many different statistical techniques ranging from data mining, predictive modeling to machine learning. Predictive modeling may be applied to many areas such as weather forecasting, business, Bayesian spam filters, advertising and marketing, fraud detection, etc. Predictive models are based on variables that are most likely to influence the outcome [22]. These variables are also known as predictors. There are many types of predictive models, such as neural networks and Decision trees. Ahmad et al. compared different methods of predictive modeling for solar thermal energy systems such as random forest, extra trees, and regression trees [23].

IBM offers a variety of services in terms of predictive analysis such as Watson analytics and SPSS [24,25,26,27,28,29,30,31,32,33,34,35,36]. AlFaris et al. reviewed the smart technologies; the interface and integration of the meters, sensors and monitoring systems with the home energy management system (HEMS) within the IoT with the outline that the smart home in practice provides the ability to the house to be net-zero energy building. Especially that it reduces the power demand and improve the energy performance by 37% better than ASHRAE standards for family villas sector [37]. Alirezaie et al. presents a framework called E-care@home, consisting of an IoT infrastructure, which provides information with an unambiguous, shared meaning across IoT devices, end-users, relatives, health and care professionals and organizations and demonstrates the proposed framework using an instantiation of a smart environment that is able to perform context recognition based on the activities and the events occurring in the home [38]. Bassoli et al. introduced a new system architecture suitable for human monitoring based on Wi-Fi connectivity, where the proposed solution lowers costs and the implementation burden by using the Internet connection that leans on standard home modem-routers, already present normally in the homes, and reducing the need for range extenders thanks to the long range of the Wi-Fi signal with energy savings of up to 91% [39]. Catherwood et al. presented an advanced Internet of Things point-of-care bio-fluid analyzer; a LoRa/Bluetooth-enabled electronic reader for biomedical strip-based diagnostics system for personalized monitoring, where practical hurdles in establishing an Internet of Medical Things network, assisting informed deployment of similar future systems are solved [40].

In this paper, the authors focused on designing a methodology for processing data obtained from sensors that measure non-electrical quantities in an SHC environment for the purpose of monitoring the presence of people in a room through the KNX (Konnex bus) and BACnet (Building Automation and Control Networks) technologies designed for SHC automation. Standard solutions using Software (SW) tools in IoT platforms are currently available and can process large amounts of measured data. These solutions monitor and optimize the quality of the indoor environment, taking into account the real needs of residents.

This paper examines the possibilities of increasing the accuracy in CO_2_ predictions in Smart Home Care (SHC) using IBM SPSS software tools in the IoT to determine occupancy times of a monitored SHC room. The accuracy of CO_2_ predictions from the processed data was compared and evaluated at daily, weekly and monthly intervals for the spring and autumn periods. To predict CO_2_ levels from the measured indoor and outdoor temperatures and relative humidity, the Radial Basis Function (RBF) method (feedforward neural network) was applied. To improve the accuracy of CO_2_ predictions, a wavelet transform was applied to remove additive noise from the predicted signal.

For the classification of prediction quality with Wavelet Transformation (WT) additive noise cancelation, a correlation analysis (correlation coefficient R), calculated MSE (Mean Squared Error), Mean Absolute Error (MAE), Euclidean distance (ED), City Block distance (CB) are used.

## 2. Description of Used Technologies in SHC

The Smart two-floor wooden house (hereafter, Smart Home; floor area of 12.1 m × 8.2 m; (Figure 1) was built as a training center of the Moravian-Silesian Wood Cluster (MSWC). The wooden house (SHC) was built to a passive standard in accordance with standards ČSN 75 0540-2 and ČSN 730540-2 (2002) [41]. The measured values for the internal climate monitoring in the living space were evaluated by selection and measurement of CO_2_, the temperature (*T*) and relative humidity (rH) in room 204 with the application of air-quality sensors Siemens QPA2062 implemented in the BACnet system [42]. The basic element of the BACnet control system is a sub-station DESIGO PX PXC100-E.D. The DESIGO PX sub-stations are controlled by a user-friendly control panel PXM 20E [43]. The BACnet technology is used for HVAC (heating, ventilating, and air conditioning) control in SHC. The process sub-station BACnet/IP DESIGO PX-PXC100 E.D. forms a foundation of the control system for the measurement of nonelectrical quantities and control of operating and technical functions in SH [44]. The communication between the individual modules is executed via a standard BACnet protocol over Ethernet. Mutual communication between the sub-stations (peer-to-peer) is supported there [45]. KNX technology is used for lighting and blinds control and the switching on/off of socket circuits [46]. Furthermore, information about the security status of the building is transferred from the Electronic Security System (ESS) using KNX technology [47]. The KNX technology is suitably integrated into the BACnet technology using an interface providing interoperability between the communication protocols [48]. The visualization of control and regulation of the operating and technical functions in the building (HVAC, blinds, lighting, etc.) and storage of the measured data in the database was created using DESIGO Insight software tools [49]. To unify the monitoring and control of various automation and electronic systems into a single environment, it is necessary to use a software tool that allows you to consolidate several types of bus systems, standards and communication, and data protocols into o single monitoring application. In our case, we took advantage of the opportunities offered by the PI System software tool (hereinafter referred to as PI) produced by OSIsoft (Figure 1) [50].

The PI System includes SW tools such as PI ProcessBook for user-friendly data readout with the ability to create an application for the visualization and monitoring of SHC resident activities (Figure 2 and Figure 3) [51,52,53].

### Visualization for the SHC Created in the PI ProcessBook Tool

Visualization of the wooden house in the PI Process Book tool is divided into several screens, which can be continuously accessed from the main screen (Figure 2). The SHC control technology is integrated on each visualization screen in accordance with the Building Management System (BMS). These screens further comprise of buttons for entry into individual rooms. After clicking on the relevant room button, a new window, containing a detailed description of the technology used in the specific room shown in the individual charts, will appear. Each technological element is illustrated in the chart, wherein the individual charts are sorted in accordance with the groups of elements used (Figure 3). The individual technology units are then displayed on separate screens and can also be viewed from the main screen. This solution was chosen because it is not possible to place all the information about the technologies implemented on one screen so that the screen remained well-arranged. The additional distribution of the technologies into individual screens will allow the user to get a better insight into what elements belong to the individual technologies and which do not anymore. Thanks to this solution selected, orientation in the enclosed charts are easier as well as the analysis of the individual quantities and actions in the building.

The visualization, monitoring, and processing of the measured values of non-electric variables, such as the measurement of temperature, humidity, and CO_2_ for monitoring the quality of the indoor environment of the selected room in the building described, are implemented using the PI System software application and SPSS IBM SW Tool (Figure 1).

## 3. Proposed Method for Creating an Optimized Model of CO_2_ Concentration Prediction

### 3.1. Implementation of Predictive Analysis Using the IBM SPSS Modeler

The IBM SPSS Modeler allows users to build models using a simplified, easy-to-use, and object-oriented user interface. The user is provided with Modelling algorithms, such as prediction, classification, segmentation, and association detection. The model results can be easily deployed and read into databases, IBM SPSS Statistics and a wide variety of other applications. Working with IBM SPSS Modeler can be divided into three basic steps:Importing the data into IBM SPSS ModelerPerforming a series of analyses on the imported dataEvaluation and exporting the data

This sequence is also known as a Datastream because the data is flowing from the source to each analysis node and then to the output. The IBM SPSS Modeler allows the users to work with multiple data streams at once. These data streams can be build and modified using the stream canvas area of the application. These streams are created by drawing diagrams of relevant data operations. IBM SPSS Modeler’s Node Palette area of the displays shows most of the available data and modeling tools. The user may perform a simple drag and drop on each item in the nodes palette to add them to the current stream. The node palette items are divided into a few main categories as follows [54]:Source: contains nodes that allow importing data into IBM SPSS Modeler from external sources such as analytic servers, databases, XML files, Microsoft Excel etc.Record Ops: includes Nodes performing operations on data records, such as selecting, merging, and appending.Field Ops: these nodes can perform operations on data fields such as filtering, deriving new fields, and determining the measurement level for given fields.Graphs: Provides nodes that can graphically represent the data from before and after modeling.Modeling: contains available modeling algorithms such as neural networks, decision trees, clustering, and data sequencing.Output: consists of nodes that can provide outputs, such as plots, charts, evaluation, etc.Export: composed of the nodes that can export the output to other applications, such as Microsoft Excel.IBM SPSS Statistics: dedicated to nodes for importing or exporting data to IBM SPSS Statistics.

Neural networks are one of the many ways to achieve predictive analysis. IBM SPSS Modeler offers multiple types of neural networks for predictive analysis. The text further describes the procedure for determining the appropriate method of predicting the course of CO_2_ concentration from the measured values taken by the indoor temperature sensor *T*_i_ (°C), (QPA 2062) in an SHC room (range 0 to 50 °C/−35 to 35 °C, accuracy ± 1K) and relative humidity rH (%), (QPA 2062) (range 0 and 100%, accuracy ±5%), outdoor temperature *T*_o_ (°C), (AP 257/22), (range: −30 … + 80 °C, resolution: 0.1 °C) using the RBF. The RBF was selected due to its higher speed of training [55]. The RBF network is a feed-forward network that requires supervised learning. Unlike multilayer perceptron’s (MLP), this network consists of only one hidden layer. Overall, there are three layers in the RBF network: the input layer, RBF layer, and the output layer. The IBM SPSS algorithm guide describes mathematical models of these layers [56] (Figure 4) as following:

**Input layer**: $J_{0}=P\;units,\;a_{0:1},\;\ldots ,a_{0:J_{0}}$ with $a_{0:j}=X_{j}$

**RBF layer**: $j_{1}\;units,\;units,\;a_{1:1},\;\ldots a_{1:j_{1}};$ with $a_{1:j}=\emptyset_{j}\left(X\right)$(1)$$\emptyset_{j}\left(X\right)=\frac{e^{\left(-\sum_{p=1}^{P}\frac{1}{2\sigma_{jp}^{2}}{\left(x_{p}-\mu_{jp}\right)}^{2}\right)}}{\sum_{j=1}^{J_{1}}e^{\left(-\sum_{p=1}^{P}\frac{1}{2\sigma_{jp}^{2}}{\left(x_{p}-\mu_{jp}\right)}^{2}\right)}}$$

**Output layer**: $j_{2}=R\;units,\;a_{I:1},\;\ldots a_{I:j_{2}}$ with $a_{I:r}=\omega_{I:r}+\sum_{j=1}^{j1}\omega_{rj}\emptyset_{j}\left(X\right)$

Where:$X^{\left(m\right)}:$ Input vectorI: Number of layers (for RBM = 2)Ji: Number of units in layer i$\emptyset_{j}\left(X^{\left(m\right)}\right)$: jth unit for input $X^{\left(m\right)},\;j=1,\;\ldots ,\;j_{1}.$µ_j_: Center of $\emptyset_{j}$σ_j_: Width of $\emptyset_{j}$$a_{i:j}^{m}$: Unit j of layer i$\omega_{rj}$: weight connecting r_th_ output unit and j_th_ hidden unit of RBF layer

The training of RBF can be divided into two stages. The first stage determines the basis function by clustering methods and the second stage determines the weights given to the basis function. SPSS measures the accuracy of neural networks by calculating the percentage of the records for which the predicted value matches the observed value. For the continues values, the accuracy is calculated by 1 minus the average of the absolute values of the predicted values minus the observed values over the maximum predicted value minus the minimum predicted value (the following formula) [56].(2)$$\mathrm{Accuracy}=\frac{1}{n}\sum_{m=1}^{M}(1-\frac{|y_{r}{}^{\left(m\right)}-{\hat{y}}_{r}{}^{\left(m\right)}|}{max_{m}\left(y_{r}{}^{\left(m\right)}\right)-min_{m}\left(y_{r}{}^{\left(m\right)}\right)}),$$

### 3.2. Signal Trend Detection Based on Wavelet Transformation

#### Materials and Methods

In this section, we introduce a method for the CO_2_ concentration prediction optimization based on the Wavelet transformation additive noise canceling. Based on the experimental results, the predicted CO_2_ trend contains glitches representing the fast change part of the signal. Such signal segments may significantly deteriorate the quality of the prediction. We propose an optimized scheme of the neural network prediction based on the Wavelet filtration appearing as a robust method due to a wide variability of the filtration settings. Such a system significantly improves the prediction system based on the neural network.

In the signal processing, we assume that each signal *y*(*t*) is composed of two essential parts, namely, they are the signal trend *T*(*t*) and a component having a stochastic character *X*(*t*) which is perceived as the signal noise and details. Based on this definition, we can use the following signal formulation (3):(3)y(t)=T(t)+X(t)

The major problem when the signal trend is being extracted is noise detection. There are many applications of the trend detection including the CO_2_ measurement. Such a signal may be influenced by the glitches which should be removed to obtain a smooth signal for further processing. The wavelet analysis represents a transformation of the signal *y*(*t*) to obtain two types of coefficients, particularly they are the wavelet and scaling coefficients. These coefficients are completely equivalent with the original CO_2_ signal. It is supposed that wavelet coefficients are related to changes along a specifically defined scale. The main idea of the signal trend detection is to perform an association of the scaling coefficients with the signal trend *T*(*x*). On the other hand, the wavelet coefficients are supposed to be associated with the signal noise, which is mainly represented by the glitches when processing the CO_2_ signal. In our analysis, we considered an uncorrelated noise, adapting the wavelet estimator to work as a kernel estimator. The advantage of such an approach is formulating an estimator based on the sampled data irregularity. In this method, we used the scaling coefficients as estimators of the signal trend. We supposed that the sampled CO_2_ observations are represented by $Y\left(t_{n}\right)$, thus, the CO_2_ estimator is given by Equation (4):(4)$$\hat{T}\left(t\right)=\overset{N-1}{\underset{n=0}{\sum }}Y\left(t_{n}\right)\int E_{J}\left(t,s\right)ds$$

Integration is done over a set of the intervals (An(s)), their union forms perform partitioning interval covering all the observations  tn, where tn∈An. Consequently, EJ is defined as Equation (5):(5)EJ(t,s)=2−J∑k∈ℤθ(2−Jt−k)θ(2−Js−k)

In this expression, θ(t) represents the scaling function. This function is defined as follows (Equation (6)):(6)θ(t)=∑k∈ℤckθ(2t−k)

The wavelet function is defined by Equation (7):(7)$$\psi \left(t\right)={\left(-1\right)}^{k}c_{1-k}\theta \left(2t-k\right)$$

The first crucial task is an appropriate selection of the mother’s wavelet for the predicted CO_2_ signal filtration. Supposing the Daubechies wavelets can well reflect morphological structure therefore, this family was used for our model. Particularly, in our approach, we used the Daubechies wavelet (Db6), with the D6 scaling function utilizing the orthogonal Daubechies coefficients.

### 3.3. Validation Ratings Used

In order to carry out the objective comparison, the following parameters were considered:

**Mean Absolute Error (MAE)** represents the estimator measuring of the difference between two continuous variables. The *MAE* is given by the following expression:(8)$$MAE=\frac{1}{n}\overset{n}{\underset{i=1}{\sum }}\left|y_{i}-\hat{y_{ı}}\right|$$

**Mean squared error (MSE)** represents the estimator measuring the average of the error squares between two signals. The *MSE* represents a risk function which corresponds with the expected value of the squared or quadratic error loss. The *MSE* is given by the following expression:(9)$$MSE\left(x_{1},{\hat{x}}_{2}\right)=\frac{1}{n}\overset{n}{\underset{i=1}{\sum }}{\left(x\left(i\right)-\hat{x}\left(i\right)\right)}^{2}$$

**Euclidean distance (ED)** represents an ordinary straight-line distance between two points lying in the Euclidean space. Based on this distance, the Euclidean space becomes a metric space. The lower the Euclidean distance we achieve, the more similar are two signal samples. In our analysis, we considered a mean of the *ED*. The Euclidean distance is given by the following expression:(10)$$d\left(x_{1},{\hat{x}}_{2}\right)=\sqrt{{\left(x_{1}-{\hat{x}}_{2}\right)}^{2}+{\left(y_{1}-{\hat{y}}_{2}\right)}^{2}}$$

**City Block distance (CB)** represents a distance between two signals $x_{1},{\hat{x}}_{2}$ in the space with the Cartesian coordinate system. This parameter can be interpreted as a sum of the lengths of the projections of the line segments between the points onto the coordinate axes. CB distance is defined as follows:(11)$$d_{cb}\left(x_{1},{\hat{x}}_{2}\right)=∥x_{1}-{\hat{x}}_{2}∥=\overset{n}{\underset{i=1}{\sum }}\left|x_{1i}-{\hat{x}}_{2i}\right|$$

**The Correlation coefficient (*R*)** measures a level of the linear dependency between two signals. The more the signals are considered linearly dependable, the higher the correlation coefficient is. In comparison with the previous parameters, the correlation coefficient represents a normative parameter. Zero correlation stands for the total dissimilarity between two signals, measured in a sense of their linear dependency. Contrarily, 1 and −1 stand for full positive and full negative correlation.

As we have already stated above, in our work, we analyze two-month CO_2_ predictions. In each measurement, we have a prediction from the neural network with 10, 50, 100, 150, 200, 250, 300, 350 and 400 neurons. Thus, we completely analyzed 9 predicted signals for each measurement. These signals are compared against the reference based on the evaluation parameters stated above. In terms of the Euclidean distance and MSE, lower values indicate a higher agreement between the signal and reference and thus, a better result. Contrarily, a higher correlation coefficient shows better results. In the following part of the analysis, we report the results of the quantification comparison. All the testing is done for the Wavelet Db6, with 6-level decomposition and the Wavelet settings as follows: threshold selection rule—Stein’s Unbiased Risk and soft thresholding for selection of the detailed coefficients.

## 4. The Practical Experimental Part

### 4.1. First Part of the ADL Information in SHC from the CO_2_ Concentration Course, Blinds, Slats and On/Off Control of Lights

The first experimental part in the study addressed the real needs of seniors who live in their own flats despite advanced age and mental and physical disabilities. These people strive to maintain maximum self-sufficiency and, thus, remove as much burden from their relatives, neighbors, friends or surroundings as possible. An example is a married couple, one of whom is mentally impaired, the other being the caregiver. They stay in touch with their family (their children) by SMS to keep them informed about how they are. In situations of acute need, the children are ready to come and help. In this example, the indirect ADL (Activities of Daily Living) in Room R203 (Figure 2) in SHC can be detected by monitoring operational and technical functions, such asturning the lights on/off (Figure 5)opening/closing windows (Figure 6)raising/lowering blinds (Figure 7)rotating blind slats (Figure 8)increasing/decreasing the CO_2_ concentration (Figure 5, Figure 6, Figure 7 and Figure 8).

#### Discussion of the First Experimental Part

In the aforementioned results of Experimental Part 1 (Figure 5, Figure 6, Figure 7 and Figure 8), the presence of persons in the SHC monitored area is clearly time-localized according to the ADL. ADL information in SHC can be thus reliably forwarded to the close relatives. The information obtained at daily, weekly and monthly intervals can also be used alongside SHC automation technologies in a so-called “smart building”. This type of building records house activities and uses the accumulated data to automatically control technologies according to the predictable needs of users, such as controlling lights, blinds, heating, forced ventilation and cooling based on the usual patterns of use and facilitates cost savings for programming and configuring the intelligent house control system [2].

### 4.2. Second Experimental Part: ADL Monitoring Information from Prediction CO_2_ Concentration Course Within IoT SPSS SW Tool

As it was described earlier, the main goal is predicting CO2 concentration based on data collected by indoor humidity, indoor temperature, and outdoor temperature. The sample data were collected from the SHC. The procedure of this implementation can be divided into a few steps as following:Pre-processing the dataDeveloping a data stream using IBM SPSS ModelerTesting various training data from different times of the yearAnalyzing the results and selecting the best modelUploading the selected model method to Watson studio for IoT implementation

#### 4.2.1. Pre-Processing

Data Normalization using the min-max method (often known as feature scaling) was used as a pre-processing method. This method scales the parameters in the range between 0 and 1. Since the experimental data were stored in data files, the pre-processing stage was performed. The implementation was performed by calculating the minimum and maximum values of each parameter. Then the normalized values were calculated using Equation (12).(12)$$Normalized\;value=\frac{current\;value-\min\;value}{\max\;value-\min\;value}$$

#### 4.2.2. Developing a Data Stream Using IBM SPSS Modeler

Figure 9 shows the data stream developed in IBM SPSS Modeler. In the first stage, the data were fed to IBM SPSS using Excel files by adding Excel node from the source category of the node palette. In the next stage, a data type selection operator was added from the field operator category. This node carried the task of setting the default target to CO_2_ concentration and default inputs (in this case inputs were used as predictors) to humidity, indoor temperature, and outdoor temperature values.

There are a few common validation methods used in IBM SPSS Modeler such as K-fold, V-fold, N-fold, and Partitioning. The IBM SPSS Modeler User manual recommends using the partitioning method for large datasets due to the faster processing time. The partitioning method randomly divides the data sets into three parts of training, testing, and validation. The ratio of this division can be selected in the software using percentages of the data set. The partition node form field operator category was added to the stream in order to divide it into three parts. The first part (40% of the data) for the training of the neural network, using target values (CO_2_) and predictors (humidity, indoor temperature, and output temperature). The second part for testing (30% of the data) the neural network by using target and predictor values. The testing partition is used for selecting the most suitable model and prevention of overfitting. The last partition was dedicated for validation (30% of the data) of the developed model using only predictors and comparing the prediction with the reference signal. In other words, the validation partition is used to determine how well the model truly performs [19].

In the next step, an automatic data preparation field operator was used in order to transform the data for better predictive accuracy. The transformed data were fed to a neural network for training a model. The neural network is using the RBF model with various numbers of neurons for multiple implementations (Figure 10). The resulting model from the trained neural network is represented by a nugget gem. Varies nodes were connected to this nugget gem for additional analysis of the model such as an Excel node for exporting reference data and predicted values to an Excel file (a filter node was used to select which values to store in the output Excel file), a Time plot node to display the time plot of reference versus predicted values, a Multiplot node for displaying the plots from portioned data and an Analysis node for displaying the details such as linear correlation, mean absolute error, etc.

#### 4.2.3. Testing Various Training Data from Different Times of the Year

For this stage of the experiment, seven different data sets from the spring and fall of 2018 were selected. The data collection was performed at the rate of one sample per minute. The first selected data interval was the whole month of May (Table 1). The validation results indicate that model number 7 with 76.6% accuracy and relevantly small error (MAE = 0.006, MSE = 9.75 × 10^−5^) represents the best prediction. By repeating the experiment with data from November 2018 (Table 2), a slight improvement in the accuracy of all models can be observed. Additionally, by observing Table 3, it is apparent that model number 9 holds the best results in terms of accuracy (80.6%), linear correlation (0.898), MAE (0.011) and MSE (1.553 × 10^−3^).

The training process was repeated by replacing the intervals with a week in May 2018 (Table 3) and a week in November 2018 (Table 4). With a reduction in the data size, the chances of overfitting were reduced, resulting in better generalization and higher accuracy (Figure 11). These improvements can be observed in Table 3 and Table 4. In both cases, the model number 9 shows the finest accuracy (6^th^ to 13^th^ of May 2018: 98.2%; 6^th^ to 13^th^ of November 2018: 96.2%), highest linear correlation (6^th^ to 13^th^ of May 2018: 0.985; 6^th^ to 13^th^ of November 2018: 0.979) and the lowest error (6^th^ to 13^th^ of May 2018: MSE: 3.14 × 10^−6^ MAE: 0.001; 6^th^ to 13^th^ of November 2018: MSE: 4.12 × 10^−5^, MAE: 0.003). Additionally, a consistency between the accuracy results of the two experiments can be observed.

For the last few experiments, the size of the data sets was reduced to one day. The 15^th^ and 12^th^ of May (Table 5 and Table 6) and the 15^th^ of November (Table 7), Figure 12 were selected for this experiment. Once again, a significant improvement of the accuracies due to the reduction in the interval lengths can be observed. The result from the 12^th^ of May (Table 5) implies that model number 5 shows the maximum accuracy (98.1%). By considering the linear correlation and error values, it can be concluded that model number 3 shows better overall characteristics (Figure 13). In the case of 15^th^ of May, model number 3 provides the most accurate result with an impressive 99.9% of accuracy, closely followed by model number 9 with 99.8% of accuracy. In the next step, the experiment was repeated with data from November 15. Similar to the few of the previous cases, model number 9 shows the highest accuracy (99.7%), relevantly high linear correlation value (0.895) and the lowest errors (MAE: 0.003, MSE: 6.45 × 10^−5^).

#### 4.2.4. Analyzing the Results and Selecting the Best Model

Table 8 shows the average of the accuracy, linear correlation MAE and MSE in each experiment. By observing this table, it is apparent that the experiments with an interval length of one month, hold the lowest average accuracies (63.1% and 72%). It can also be observed that the experiment with the 15^th^ of May as period holds the highest average accuracy (99.5%), highest linear correlation (0.995) and relevantly low error values (MAE: 1.78 × 10^−3^, MSE: 2.44 × 10^−5^). Additionally, the experiment with a interval length of a week in May shows a slightly smaller average accuracy (94.7%) and linear correlation (0.967) but it has overall lower error values (MAE: 1.78 × 10^−3^, MSE: 2.44 × 10^−5^) (Figure 14).

Table 9 contains the average results from all experiments for each model. The overall trend of this Table points toward the fact that with the increasing number of neurons, the accuracy of the model’s increases. This increase may be insignificant after a certain point. For example, models number 5, 6, 7, 8 and 9 share a similar average accuracy (91.2%, 91.1%, 90.2%, 90.2% and 92.1%), linear correlation (0.935, 0.931, 0.935, 0.936 and 0.936), MAE (8.15 × 10^−3^, 9.14 × 10^−3^, 7.71 × 10^−3^, 7.71 × 10^−3^ and 7.85 × 10^−3^) and MSE (1.02 × 10^−3^, 1.45 × 10^−3^, 1.15 × 10^−3^, 1.15 × 10^−3^ and 1.17 × 10^−3^).

Figure 11 clearly demonstrates the relationship between accuracy, the experiments interval length and the number of neurons. By summing up all of the obtained results, it is clear that model number 3 (Table 6) trained with the data from 15^th^ of May holds the highest accuracy (99.8%), linear correlation (0.996) and relevantly small error values. Model number 9 shows the best overall average accuracy (Table 9) and, in case of the experiments with the interval of 12^th^ (see Figure 12) and 15^th^ May, the accuracy difference between this model and the most accurate model (model number 3) is negligible. Therefore, model number 9 trained with the May 15^th^ interval was selected as the most suitable model for the next stages (Figure 12).

#### 4.2.5. IoT Implementation with Watson Studio SW Tool

The data streams created in the IBM SPSS Modelers can be stored as a file (“.str” format). The IBM Watson Studio allows the user to import the developed data stream simply by uploading the stored files. This allows the data streams that were originally developed in the IBM SPSS Modeler to take advantage of cloud computing, cloud storage and the possibility of near-real-time streaming. As it was explained earlier, model number 9 (with 400 neurons) trained with the data from 15th of May was selected as the best overall result of this experiment. Specifically, this model showed high accuracy, high linear correlation, low MAE and MSE errors. Therefore, it was uploaded to Watson studio. Figure 15 shows the streamed developed in SPSS in Watson studio for near-real-time training (Excel files were replaced with assets on the cloud). Figure 16 shows a data flow stream in Watson that includes model 9 trained with data from the 15^th^ of May for near real-time prediction.

#### 4.2.6. Discussion of the Second Experimental Part

By evaluating the obtained results from the implementation with IBM SPSS Modeler, it is apparent that as it was expected that the experiments with interval length of one day showed better overall accuracy (average value up to 99.5%) and the experiments with sample periods of one month showed the least overall accuracy (average value up to 72.2%). Additionally, in four out of six experiments, model number 9 held the highest accuracy, and in the other two cases, it had the only insignificant difference with the most accurate models. Therefore, it was selected as the overall most accurate model. As it was mentioned earlier, in terms of the training interval, 15^th^ of May showed the most accurate results. Therefore, model number 9 with the 15^th^ of May training interval (Table 9) was selected and exported to an IBM cloud data stream. Furthermore, the results demanded additional filtering in order to reduce the noise and provide smoother results.

### 4.3. Third Part—Testing and Quantitative Comparison (WT Additive Noise Canceling)

#### 4.3.1. Testing and Quantitative Comparison

In our research, we analyzed signals representing the CO_2_ signals. We had a set of the estimated (predicted) signals being compared against the real measured signal CO_2_, which is perceived as a reference. In our analysis, we are comparing two-month CO_2_ prediction. We compared one-day, one-week and one-month predictions for May and November 2018.

Based on the observations, it is apparent that the predicted CO_2_ signals do not have a smooth process. They are frequently influenced by rapid oscillations, so-called glitches and signal fluctuations (Figure 17 and Figure 18). Such signal variations represent the signal noise, impairing the real trend of the CO_2_ prediction, which should be reduced. In our analysis, we used wavelet filtration to eliminate such signals to obtain the signal trend for further processing.

As we have already stated above, we used the mother’s wavelet Db6 for the CO_2_ signal trend detection. Firstly, we take advantage of the fact that different level of the decomposition allows perceiving more or less signal details represented by the detailed coefficients. Since we need to perceive the signal trend by eliminating the steep fluctuations, we need to consider an appropriate level of the decomposition. An experimental comparison of individual wavelet settings is reported in Figure 17 and Figure 18. Based on the experimental results, we used the 6-level decomposition for the CO_2_ signal trend detection. The filtration procedure further utilizes the following settings: threshold selection rule—Stein’s Unbiased Risk and soft thresholding for selection of the detailed coefficients.

Wavelet filtration was used for the extraction of the CO_2_ signal trend, simultaneously rapid changes of the signal were removed. On Figure 19 and Figure 20, there is a comparison among the reference signal and predicted signals by wavelet transformation for day and month predictions from May and November 2018.

Based on the results, wavelet filtration is capable of filtering rapid signal changes whilst preserving the signal trend. To justify this situation, we report the selected situations showing the glitches deteriorating a smooth signal trend, and a respective wavelet approximation largely reducing such signal parts (Figure 21, Figure 22 and Figure 23). Among these cases, we mark the most significant glitches as green in the originally predicted signals to highlight the Wavelet smoothing effectivity.

As it is obvious, the CO_2_ prediction contains lots of significant occurrences represented by the glitches and spikes, significantly deteriorating the smoothness of the analyzed signal. Wavelet appears to be a reliable alternative for reduction of those parts of the signal. On the other hand, we are aware that trend detection, in some cases, reduces the peaks and thus, the original signal’s amplitude is reduced. Such situations are reported in Figure 21, Figure 22 and Figure 23.

In the last part of our analysis, the objective comparison is carried out. As we have already stated, we are comparing predicted CO_2_ signals with signals being filtered out by the wavelet transformation. All the signals are compared against the reference CO_2_ signals for day and week predictions.

As we have already stated above, in our work we analyzed the two-month CO2 prediction. In each measurement, we have a prediction from a neural network with 10, 50, 100, 150, 200, 250, 300, 350 and 400 neurons. Thus, we completely analyzed the predicted signals of 9 models for each measurement. These signals are compared against the reference based on the evaluation parameters stated above. In terms of the Euclidean distance and MSE, lower values indicate a higher agreement between the signal and reference and thus, better result. Contrarily, a higher correlation coefficient indicates better results. In the following part of the analysis, we report the results of the quantification comparison. All the testing is done for the Wavelet Db6, with 6-level decomposition and the following Wavelet settings: threshold selection rule—Stein’s Unbiased Risk and soft thresholding for selection of the detailed coefficients. Figure 24 shows the MSE evaluation for CO_2_ prediction in the following time intervals: (a) 6–13 May 2018, (b) 15 May 2018, (c) 1–29 May 2018, (d) 6–13 November 2018, (e) 15 November 2018, (f) 4 November–3 December 2018. Figure 25 shows the correlation coefficient evaluation for the CO_2_ prediction in the following time intervals: (a) 6–13 May 2018, (b) 15 May 2018, (c) 1–29 May 2018, (d) 6–13 November 2018, (e) 15 November 2018, (f) 4 November–3 December 2018. Figure 26 shows the Euclidean distance evaluation for the CO_2_ prediction in the following time intervals: (a) 6–13 May 2018, (b) 15 May 2018, (c) 1–29 May 2018, (d) 6–13 November 2018, (e) 15 November 2018, (f) 4 November–3 December 2018.

Lastly, we summarize the achieved results of all the predicted signals. In Table 10 and Table 11, we compare the individual parameters—Mean Square Error (MSE), Correlation index (Corr) and Euclidean distance (ED)—for individual CO_2_ predictions from May and November 2018. Each parameter is averaged for all the predictions and the difference (diff) between the original prediction and Wavelet smoothing is evaluated (Table 12).

#### 4.3.2. Discussion of the Third Experimental Part

As it is obvious, the predicted CO_2_ signals contain lots of significant occurrences represented by the glitches and spikes, significantly deteriorating the smoothness of the analyzed signals. Such steep fluctuations may have a significant impact on CO_2_ accuracy. Wavelet appears to be a reliable alternative for reduction of those parts of the signal. On the other hand, we are aware that trend detection, in some cases, reduces the peaks and thus, the original signal’s amplitude is reduced. In our work, we have studied the Daubechies wavelet family. These wavelets, as it is known, can well reflect the morphological structure of the signals. We are particularly using the Db6 wavelet for trend detection.

Alternatively, we mention the comparison in Reference [50] of the CO_2_ filtration based on the LMS algorithm. In this study, the authors employed adaptive filtration. The main limitation of this method is a necessity of the reference signal and a slow adaptation of the filtration procedure, as well as depending on the accuracy of the step size parameter μ calculation and inaccurate determination of the arrival and departure time of the person from the monitored area. Furthermore, the Wavelet filtration presented in this study achieves better results in a context of the objective comparison against the LMS filtration. Wavelet filtration has a much stronger potential for the CO_2_ filtration due to a possibility of the application of a variety wavelets allowing for the extraction of specific morphological signal features in various decomposition levels and, thus, better optimize the CO_2_ prediction. These facts predetermine wavelets to be a robust system for the CO_2_ prediction enhancement.

In the last part of our analysis, the objective comparison is carried out. As we have already stated, we compared originally measured CO_2_ signals with predicted signals being filtered out by the wavelets. To carry out the objective comparison, the following parameters are considered: when considering the MSE, we get better results for wavelet trend detection. This means that we have minimized the difference between the gold standard and filtered signals. The correlation coefficient gives higher values for the predicted CO_2_ signals. Regardless, we have achieved just slight differences. The reason might be caused by the fact that the trend detection largely omits higher peaks, therefore, the linear dependence for wavelet filtration is smaller when compared with the predicted signals. Using the Wavelet filtration leads to more accurate results against the predicted signals and signals are much more smoothed, not containing steep fluctuations. On the other hand, we are aware of a certain loss of the amplitude. Therefore, in the future, it would be worth investigating the frequency features of the CO_2_ signals to objectively determine frequency modifications while filtering by the wavelets.

## 5. Conclusions

The authors of the paper focused on designing a methodology that specifies a procedure for processing data measured by sensors in an SHC environment for the purpose of indirectly monitoring the presence of people in an SHC area through KNX and BACnet technologies commonly applied in building automation. This paper explores the possibilities of improving accuracy in CO2 predictions in SHC using IBM SPSS software tools in the IoT to determine the occupancy times of a monitored SHC room. The RBF method was applied to predict CO_2_ levels from the measured indoor and outdoor temperatures and relative humidity. The accuracy of CO_2_ predictions from the processed data was compared and evaluated at daily, weekly and monthly intervals for the spring and autumn periods. As it was expected the most accurate results were provided by experiments with the daily intervals (accuracy was about 99%) while the monthly intervals resulted in the least accurate results (accuracy was about 80%). Overall, the developed stream in IBM SPSS Modeler is capable of predicting the CO_2_ concentration values using the values of humidity and indoor and outdoor temperature. By providing a live data asset to the IBM Cloud, the uploaded model can achieve near real-time prediction of CO_2_ concentration values. Using a wavelet transform mathematical method to cancel additive noise led to more accurate results in predicted signals. The signals were also much smoother and did not contain sharp fluctuations, although there was a certain loss in amplitude, resulting in inaccuracies when the maximum achieved CO_2_ value was determined. Future work should, therefore, focus on finding an optimal method for canceling additive noise in real time, which would help increase the overall accuracy of CO_2_ predictions [57,58,59,60]. Additionally, the real-life and live performance of this implementation should be examined.

## Figures and Tables

**Figure 1 sensors-19-01407-f001:**
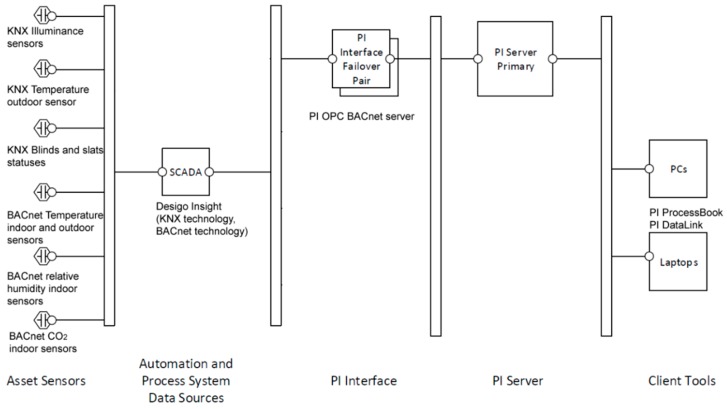
A block diagram of data transfer from the Smart Home Care technology through the PI OPC (Ole for Process Control) interface to the PI Server and PI ProcessBook.

**Figure 2 sensors-19-01407-f002:**
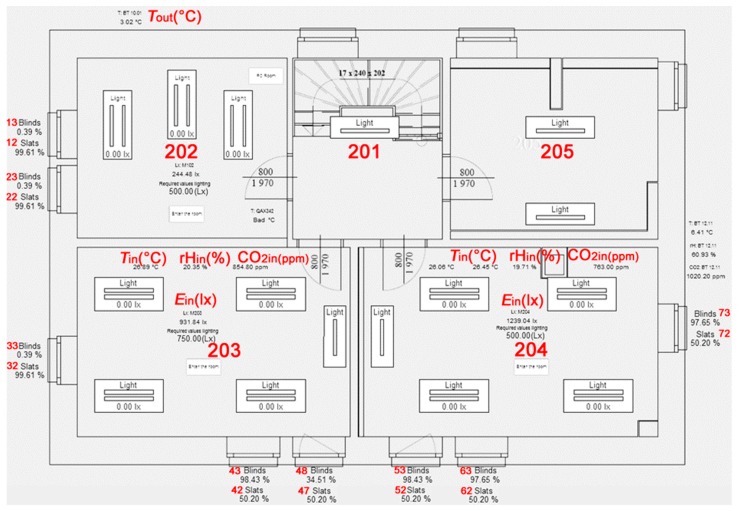
The main visualization screen in SW tool PI ProcessBook, first floor.

**Figure 3 sensors-19-01407-f003:**
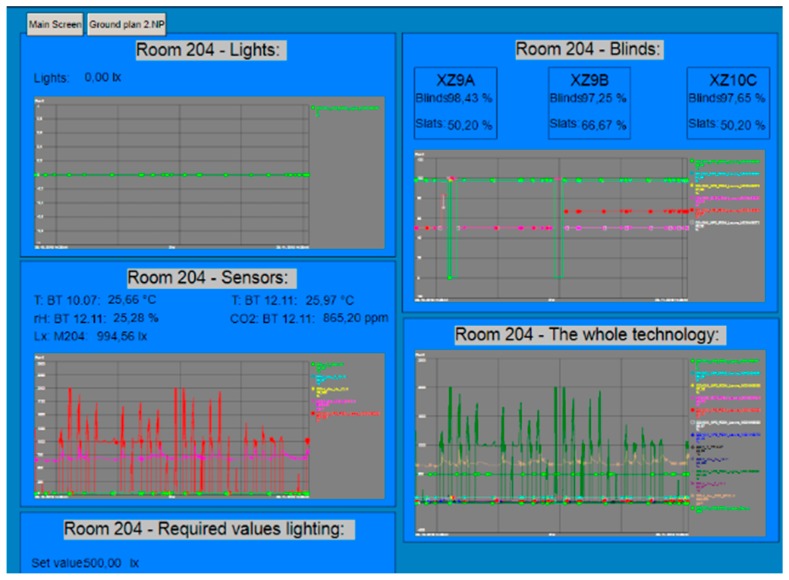
A screen for room 204 with a detailed description of individual technologies and individual charts in SW tool PI ProcessBook.

**Figure 4 sensors-19-01407-f004:**
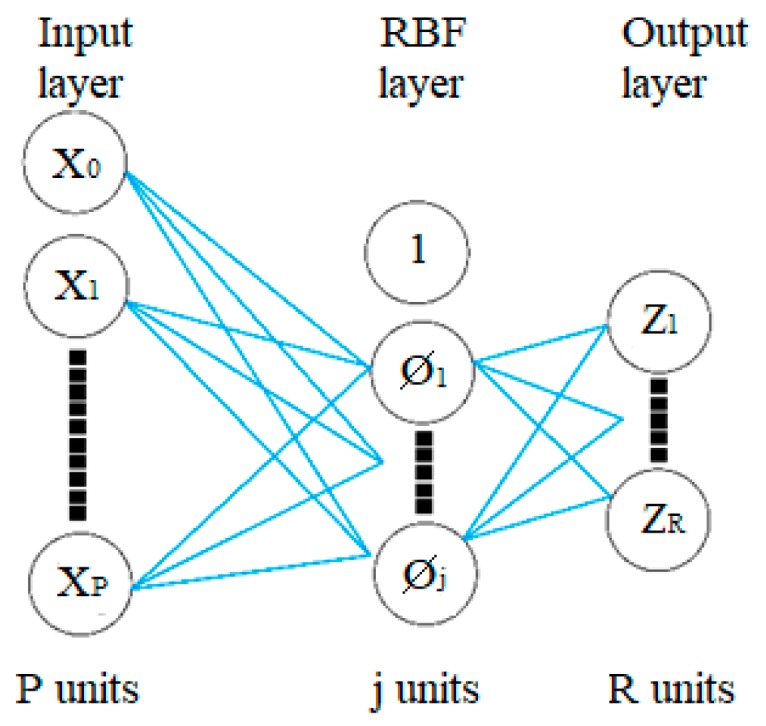
The Radial Basis Function (RBF) neural network diagram.

**Figure 5 sensors-19-01407-f005:**
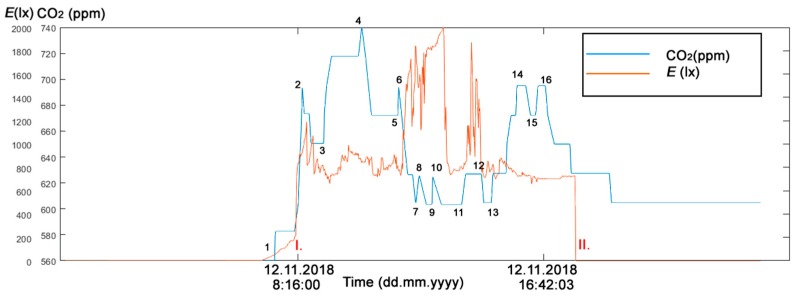
The reference (measured) CO_2_ concentration course and Illuminance *E* (lx) course (12 November 2018) in room R203 in Smart Home Care (SHC) for Activities of Daily Living (ADL) monitoring. (1) arrival (12 November 2018 7:33:00), (2) departure (12 November 2018 8:28:11), Time of Person Presence (TPP) ∆*t*_1_ = 0:55:11; (3) arrival (12 November 2018 09:12:52), (4) departure (12 November 2018 10:29:16), TPP ∆*t*_2_ = 1:16:24; (5) arrival (12 November 2018 11:41:55), (6) departure (12 November 2018 11:44:18), TPP ∆*t*_3_ = 0:02:23. (7) arrival (12 November 2018 12:18:04), (8) departure (12 November 2018 12:26:15), TPP ∆*t*_1_ = 0:08:11; (9) arrival (12 November 2018 12:51:50), (10) departure (12 November 2018 12:53:12), TPP ∆*t*_2_ = 0:01:22; (11) arrival (12 November 2018 13:51:52), (12) departure (12 November 2018 14:32:47), TPP ∆*t*_3_ = 0:40:55; (13) arrival (12 November 2018 14:52:34), (14) departure (12 November 2018 16:02:30), TPP ∆*t*_2_ = 1:09:56; (15) arrival (12 November 2018 16:23:18), (16) departure (12 November 2018 16:42:03), TPP ∆*t*_3_ = 0:18:45. Light on I. 8:15:54, Light off II. 17:43:47.

**Figure 6 sensors-19-01407-f006:**
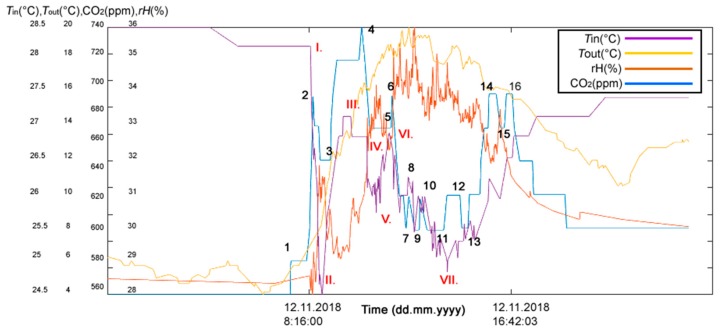
The reference (measured) CO_2_ concentration course, indoor temperature and relative humidity course (12 November 2018), in room R203, outdoor temperature course (12 November 2018) within the measurement of nonelectrical values in SHC for ADL monitoring. (1) arrival (12 November 2018 7:33:00), (2) departure (12 November 2018 8:28:11), (∆*t*_1_ = 0:55:11; (3) arrival (12 November 2018 09:12:52), (4) departure (12 November 2018 10:29:16), (TPP ∆*t*_2_ = 1:16:24; (5) arrival (12 November 2018 11:41:55), (6) departure (12 November 2018 11:44:18), (TPP ∆*t*_3_ = 0:02:23. (7) arrival (12 November 2018 12:18:04), (8) departure (12 November 2018 12:26:15), (TPP ∆*t*_1_ = 0:08:11; (9) arrival (12 November 2018 12:51:50), (10) departure (12 November 2018 12:53:12), (TPP ∆*t*_2_ = 0:01:22; (11) arrival (12 November 2018 13:51:52), (12) departure (12 November 2018 14:32:47), (TPP ∆*t*_3_ = 0:40:55; (13) arrival (12 November 2018 14:52:34), (14) departure (12 November 2018 16:02:30), (TPP ∆*t*_2_ = 1:09:56; (15) arrival (12 November 2018 16:23:18), (16) departure (12 November 2018 16:42:03), TPP ∆*t*_3_ = 0:18:45. I. The window was opened at 8:22:44, II. The window was closed at 8:50:42, III. The window was opened partially at 10:02:19, IV. The entire window was opened at 10:42:34, V. The window was closed at 11:05:25, VI. The window was opened partially at 11:36:07, VII. The window was closed at 14:00:35.

**Figure 7 sensors-19-01407-f007:**
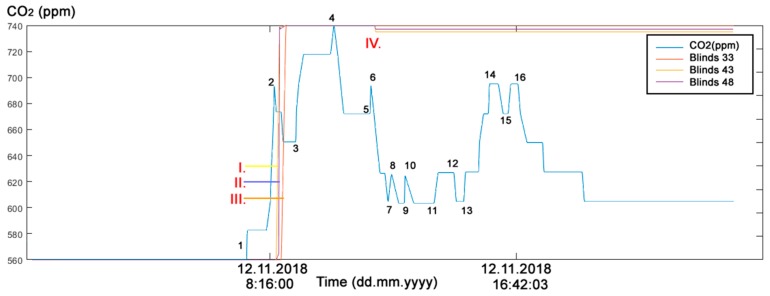
The reference (measured) CO_2_ concentration course and Blinds course (12 November 2018) in room R203 in SHC for ADL monitoring. (1) arrival (12 November 2018 7:33:00), (2) departure (12 November 2018 8:28:11), ∆*t*_1_ = 0:55:11; (3) arrival (12 November 2018 09:12:52), (4) departure (12 November 2018 10:29:16), TPP ∆*t*_2_ = 1:16:24; (5) arrival (12 November 2018 11:41:55), (6) departure (12 November 2018 11:44:18), TPP ∆*t*_3_ = 0:02:23. (7) arrival (12 November 2018 12:18:04), (8) departure (12 November 2018 12:26:15), TPP ∆*t*_1_ = 0:08:11; (9) arrival (12 November 2018 12:51:50), (10) departure (12 November 2018 12:53:12), TPP ∆*t*_2_ = 0:01:22; (11) arrival (12 November 2018 13:51:52), (12) departure (12 November 2018 14:32:47), TPP ∆*t*_3_ = 0:40:55; (13) arrival (12 November 2018 14:52:34), (14) departure (12 November 2018 16:02:30), TPP ∆*t*_2_ = 1:09:56; (15) arrival (12 November 2018 16:23:18), (16) departure (12 November 2018 16:42:03), TPP ∆*t*_3_ = 0:18:45. I. The Blind 43 (Figure 2) was lowered down in time 8:32:37, II. The Blind 48 (Figure 2) was lowered down in time 8:37:24 III. The Blind 33 (Figure 2) was lowered down in time 8:43:12, IV. The blinds were pulled partially in time 11:52:29.

**Figure 8 sensors-19-01407-f008:**
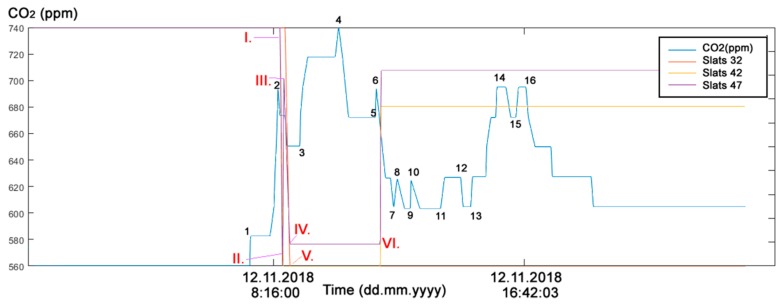
The reference (measured) CO_2_ concentration course and Slats course (12 November 2018) in room R203 (Figure 2) in SHC for ADL monitoring. (1) arrival (12 November 2018 7:33:00), (2) departure (12 November 2018 8:28:11), ∆*t*_1_ = 0:55:11; (3) arrival (12 November 2018 09:12:52), (4) departure (12 November 2018 10:29:16), TPP ∆*t*_2_ = 1:16:24; (5) arrival (12 November 2018 11:41:55), (6) departure (12 November 2018 11:44:18), TPP ∆*t*_3_ = 0:02:23. (7) arrival (12 November 2018 12:18:04), (8) departure (12 November 2018 12:26:15), (TPP ∆*t*_1_ = 0:08:11; (9) arrival (12 November 2018 12:51:50), (10) departure (12 November 2018 12:53:12), (TPP ∆*t*_2_ = 0:01:22; (11) arrival (12 November 2018 13:51:52), (12) departure (12 November 2018 14:32:47), TPP ∆*t*_3_ = 0:40:55; (13) arrival (12 November 2018 14:52:34), (14) departure (12 November 2018 16:02:30), TPP ∆*t*_2_ = 1:09:56; (15) arrival (12 November 2018 16:23:18), (16) departure (12 November 2018 16:42:03), TPP ∆*t*_3_ = 0:18:45.I. Slats 47 (Figure 2) were turned at 8:32:37, II. Slats 42 were turned at 8:33:18. III. Slats 42 were turned at 8:38:05, IV. Slats 42 were turned at 8:39:47, V. Slats 32 were turned at 8:43:12, IV. Slats 42 and 47 were turned at 11:52:29.

**Figure 9 sensors-19-01407-f009:**
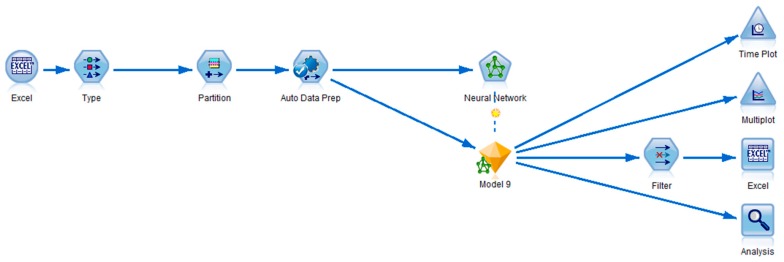
The developed stream using IBM SPSS Modeler.

**Figure 10 sensors-19-01407-f010:**
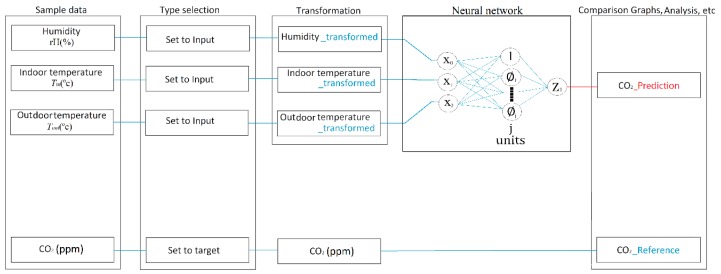
The block diagram of the developed Stream using IBM SPSS Modeler.

**Figure 11 sensors-19-01407-f011:**
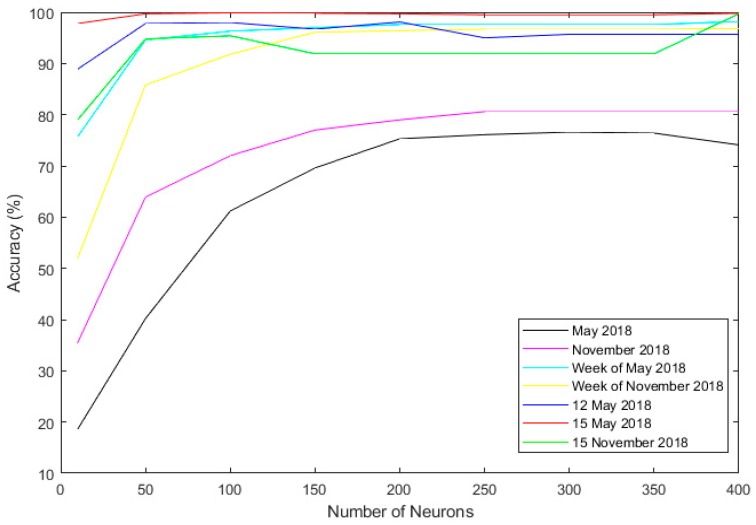
The accuracy for each interval of experiments.

**Figure 12 sensors-19-01407-f012:**
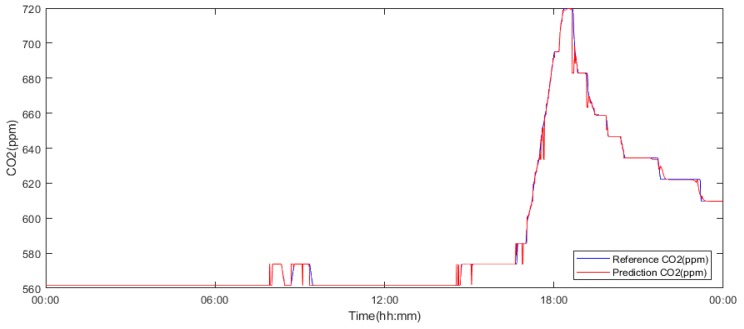
Model number 9 trained and validated using data from yjr 15^th^ of May 2018.

**Figure 13 sensors-19-01407-f013:**
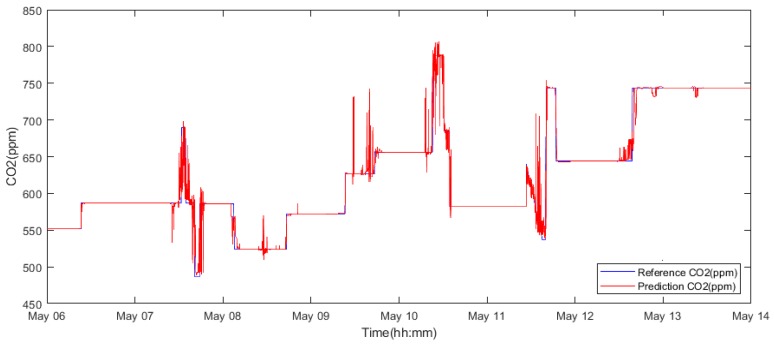
Model number 9 trained and validated using data from the interval of the 6^th^ of May 2018 to 13^th^ of May 2018.

**Figure 14 sensors-19-01407-f014:**
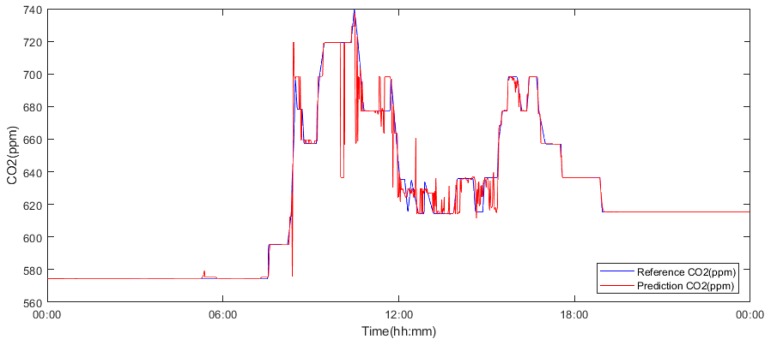
Model number 3 trained and validated using data from the interval of 12^th^ of May 2018.

**Figure 15 sensors-19-01407-f015:**
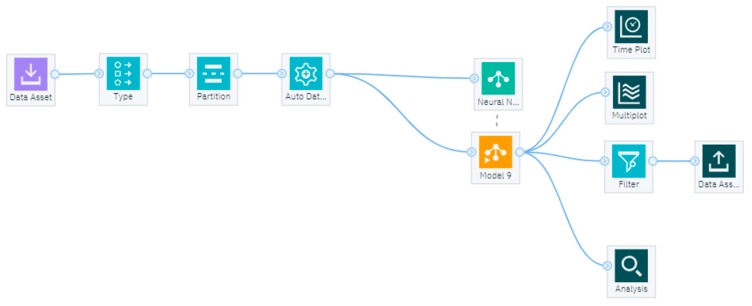
Watson Studio with the data stream developed in IBM SPSS Modeler.

**Figure 16 sensors-19-01407-f016:**
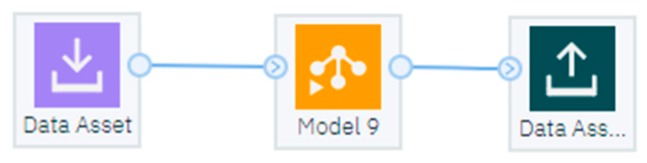
Watson Studio with the data flow stream that uses model 9.

**Figure 17 sensors-19-01407-f017:**
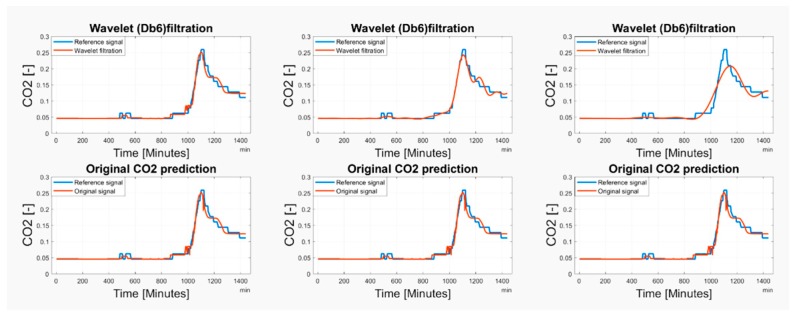
The comparison of the original predicted CO_2_ signal and Wavelet filtration from 15 May 2018 for 2-level decomposition (**left**), 6-level decomposition (**middle**) and 7-level decomposition (**right**).

**Figure 18 sensors-19-01407-f018:**
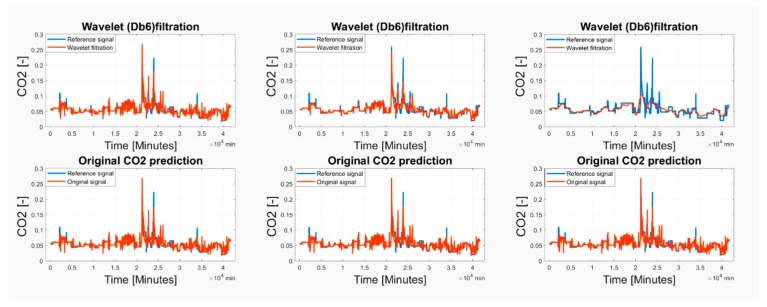
The comparison of the original predicted CO_2_ signal and Wavelet filtration from 1–29 May 2018 for 2-level decomposition (**left**), 6-level decomposition (**middle**) and 7-level decomposition (**right**).

**Figure 19 sensors-19-01407-f019:**
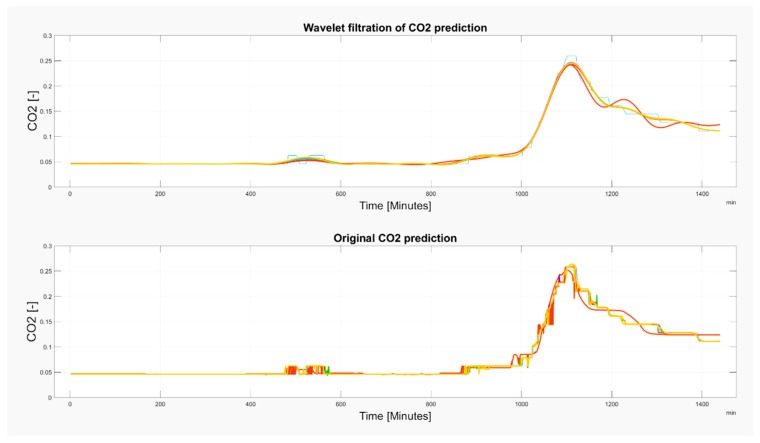
An example of one-day prediction from 15^th^ May 2018 of the CO_2_ signal. Filtration is done by using the Db6 wavelet with 6-level decomposition.

**Figure 20 sensors-19-01407-f020:**
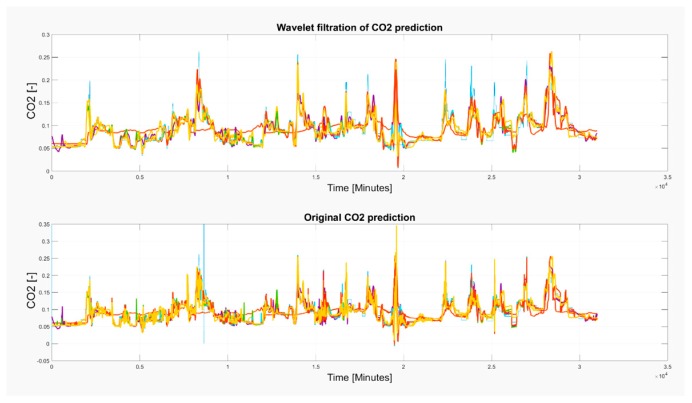
An example of whole-month prediction from November 2018 of the CO_2_ signal. Filtration is done by using the Db6 wavelet with 6-level decomposition.

**Figure 21 sensors-19-01407-f021:**
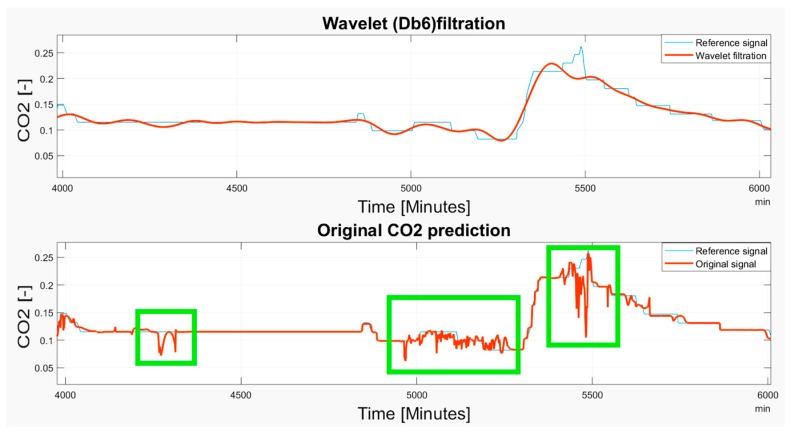
The 6–13 November 2018 prediction of the CO_2_ signal containing several glitches and signal spikes, marked as green rectangle.

**Figure 22 sensors-19-01407-f022:**
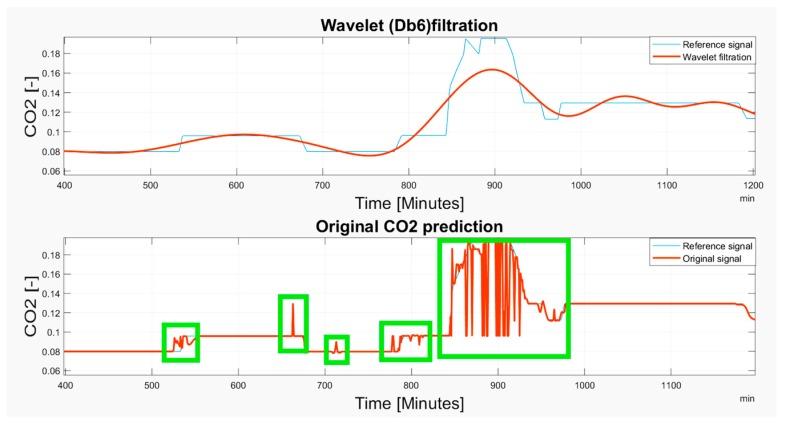
The 15 November 2018 prediction of the CO_2_ signal containing several glitches and signal spikes, marked as green rectangle.

**Figure 23 sensors-19-01407-f023:**
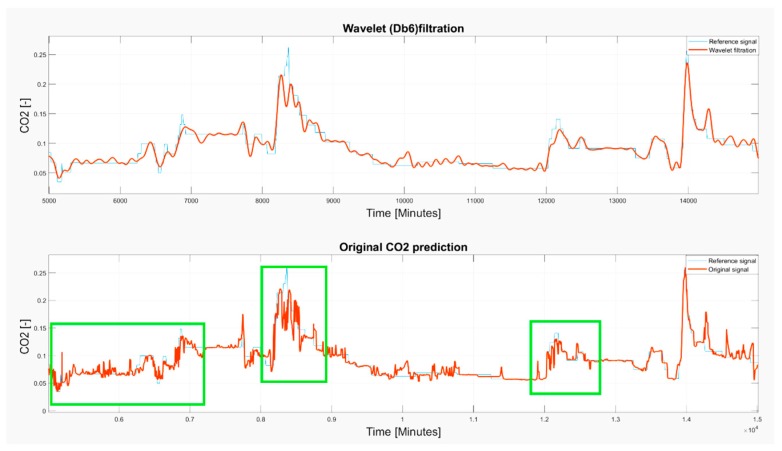
The 4 November–3 December 2018 prediction of the CO_2_ signal containing several glitches and signal spikes, marked as green rectangle.

**Figure 24 sensors-19-01407-f024:**
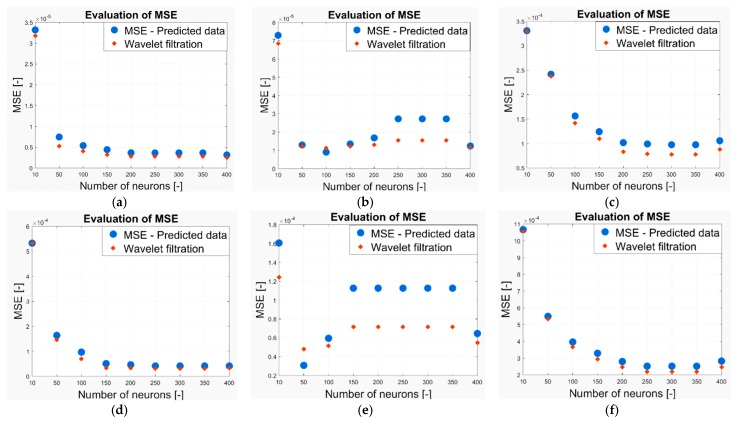
The Mean Squared Error (MSE) evaluation for the CO_2_ prediction (**a**) 6–13 May 2018, (**b**) 15 May 2018, (**c**) 1–29 May 2018, (**d**) 6–13 November 2018, (**e**) 15 November 2018, (**f**) 4 November–3 December 2018.

**Figure 25 sensors-19-01407-f025:**
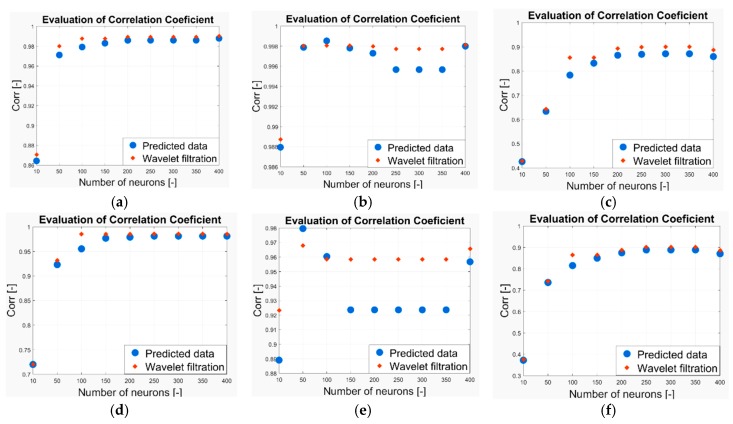
The correlation coefficient evaluation for the CO_2_ prediction (**a**) 6–13 May 2018, (**b**) 15 May 2018, (**c**) 1–29 May 2018, (**d**) 6–13 November 2018, (**e**) 15 November 2018, (**f**) 4 November–3 December 2018.

**Figure 26 sensors-19-01407-f026:**
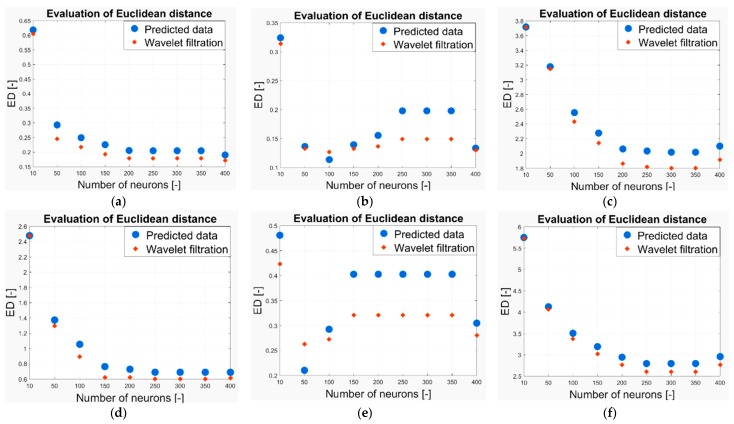
The Euclidean distance evaluation for CO_2_ prediction (**a**) 6–13 May 2018, (**b**) 15 May 2018, (**c**) 1–29 May 2018, (**d**) 6–13 November 2018, (**e**) 15 November 2018, (**f**) 4 November–3 December 2018.

**Table 1 sensors-19-01407-t001:** The trained models—Training period: May 2018.

Order of Measurement (Model Number)	Number of Neurons (-)	Accuracy (%)	Testing		Validation		
			Linear Correlation	MAE	Linear Correlation	MAE	MSE
1	10	18.6	0.405	0.012	0.432	0.012	3.31 × 10^−4^
2	50	40.2	0.628	0.010	0.638	0.010	2.42 × 10^−4^
3	100	61.2	0.782	0.008	0.787	0.008	1.56 × 10^−4^
4	150	69.6	0.828	0.007	0.831	0.007	1.24 × 10^−4^
5	200	75.3	0.859	0.006	0.864	0.006	1.02 × 10^−4^
6	250	76.1	0.861	0.005	0.867	0.006	9.91 × 10^−5^
**7**	**300**	**76.6**	**0.864**	**0.005**	**0.870**	**0.006**	**9.75 × 10^−5^**
8	350	76.4	0.864	0.005	0.870	0.006	9.75 × 10^−5^
9	400	74.1	0.854	0.006	0.861	0.006	1.06 × 10^−4^

**Table 2 sensors-19-01407-t002:** The trained models—Training period: November 2018.

Order of Measurement (Model Number)	Number of Neurons (-)	Accuracy (%)	Testing		Validation		
			Linear Correlation	MAE	Linear Correlation	MAE	MSE
1	10	35.5	0.603	0.024	0.621	0.024	1.862 × 10^−3^
2	50	63.9	0.796	0.016	0.804	0.016	1.377 × 10^−3^
3	100	72.0	0.841	0.013	0.852	0.013	1.295 × 10^−3^
4	150	77.0	0.872	0.012	0.878	0.012	1.414 × 10^−3^
5	200	79.0	0.884	0.010	0.889	0.011	1.393 × 10^−3^
6	250	80.6	0.894	0.010	0.898	0.010	1.385 × 10^−3^
7	300	80.6	0.895	0.010	0.898	0.010	1.385 × 10^−3^
8	350	80.6	0.895	0.010	0.898	0.010	1.385 × 10^−3^
**9**	**400**	**80.6**	**0.893**	**0.011**	**0.898**	**0.011**	**1.553 × 10^−3^**

**Table 3 sensors-19-01407-t003:** The trained models—6^th^ of May 2018 to 13^th^ of May 2018.

Order of Measurement (Model Number)	Number of Neurons (-)	Accuracy (%)	Testing		Validation		
			Linear Correlation	MAE	Linear Correlation	MAE	MSE
1	10	75.7	0.855	0.004	0.857	0.004	3.32 × 10^−5^
2	50	94.6	0.969	0.001	0.969	0.001	7.46 × 10^−6^
3	100	96.3	0.976	0.001	0.976	0.001	5.40 × 10^−6^
4	150	97.0	0.979	0.001	0.981	0.001	4.41 × 10^−6^
5	200	97.6	0.982	0.001	0.984	0.001	3.67 × 10^−6^
6	250	97.6	0.983	0.001	0.984	0.001	3.64 × 10^−6^
7	300	97.6	0.983	0.001	0.984	0.001	3.64 × 10^−6^
8	350	97.6	0.983	0.001	0.984	0.001	3.16 × 10^−6^
**9**	**400**	**98.2**	**0.983**	**0.001**	**0.985**	**0.001**	**3.14 × 10^−6^**

**Table 4 sensors-19-01407-t004:** The trained models—6^th^ of November 2018 to 13^th^ of November 2018.

Order of Measurement (Model Number)	Number of Neurons (-)	Accuracy (%)	Testing		Validation		
			Linear Correlation	MAE	Linear Correlation	MAE	MSE
1	10	52.0	0.724	0.016	0.714	0.017	5.33 × 10^−4^
2	50	85.8	0.915	0.008	0.921	0.008	1.64 × 10^−4^
3	100	91.8	0.947	0.005	0.954	0.005	9.68 × 10^−5^
4	150	96.1	0.969	0.003	0.973	0.003	5.07 × 10^−5^
5	200	96.4	0.973	0.003	0.977	0.003	4.61 × 10^−5^
6	250	96.7	0.978	0.003	0.978	0.002	4.13 × 10^−5^
7	300	96.7	0.978	0.003	0.978	0.002	4.13 × 10^−5^
8	350	96.7	0.978	0.003	0.978	0.002	4.13 × 10^−5^
**9**	**400**	**96.8**	**0.975**	**0.003**	**0.979**	**0.003**	**4.12 × 10^−5^**

**Table 5 sensors-19-01407-t005:** The trained models—Training period: 12^th^ of May 2018.

Order of Measurement (Model Number)	Number of Neurons (-)	Accuracy (%)	Testing		Validation		
			Linear Correlation	MAE	Linear Correlation	MAE	MSE
1	10	88.9	0.927	0.064	0.922	0.067	9.81 × 10^−3^
2	50	97.8	0.972	0.026	0.977	0.024	2.63 × 10^−3^
**3**	**100**	**98.0**	**0.971**	**0.022**	**0.967**	**0.021**	**3.11 × 10^−3^**
4	150	96.7	0.940	0.037	0.934	0.038	6.54 × 10^−3^
**5**	**200**	**98.1**	**0.937**	**0.035**	**0.943**	**0.032**	**5.43 × 10^−3^**
6	250	95.0	0.921	0.035	0.901	0.040	8.48 × 10^−3^
7	300	95.7	0.941	0.031	0.932	0.030	6.42 × 10^−3^
8	350	95.7	0.941	0.031	0.935	0.030	6.42 × 10^−3^
9	400	95.7	0.941	0.031	0.935	0.030	6.42 × 10^−3^

**Table 6 sensors-19-01407-t006:** The trained models—Training period: 15^th^ of May 2018.

Order of Measurement (Model Number)	Number of Neurons (-)	Accuracy (%)	Testing		Validation		
			Linear Correlation	MAE	Linear Correlation	MAE	MSE
1	10	97.8	0.987	0.006	0.987	0.005	7.30 × 10^−5^
2	50	99.7	0.996	0.002	0.998	0.001	1.29 × 10^−5^
**3**	**100**	**99.9**	**0.997**	**0.001**	**0.998**	**0.001**	**8.98 × 10^−6^**
4	150	99.8	0.996	0.002	0.998	0.001	1.35 × 10^−5^
5	200	99.7	0.995	0.002	0.997	0.001	1.68 × 10^−5^
6	250	99.5	0.993	0.002	0.993	0.002	2.72 × 10^−5^
7	300	99.5	0.993	0.002	0.993	0.002	2.72 × 10^−5^
8	350	99.5	0.993	0.002	0.993	0.002	2.72 × 10^−5^
9	400	99.8	0.996	0.001	0.997	0.001	1.24 × 10^−5^

**Table 7 sensors-19-01407-t007:** The trained models—Training period: 15^th^ of November 2018.

Order of Measurement (Model Number)	Number of Neurons (-)	Accuracy (%)	Testing		Validation		
			Linear Correlation	MAE	Linear Correlation	MAE	MSE
1	10	79.0	0.895	0.006	0.890	0.007	1.61 × 10^−4^
**2**	**50**	**94.8**	**0.987**	**0.002**	**0.987**	**0.002**	**3.07 × 10^−5^**
3	100	95.4	0.926	0.002	0.952	0.002	4.94 × 10^−5^
4	150	91.9	0.861	0.003	0.897	0.003	1.13 × 10^−4^
5	200	91.9	0.861	0.003	0.897	0.003	1.13 × 10^−4^
6	250	91.9	0.861	0.003	0.897	0.003	1.13 × 10^−4^
7	300	91.9	0.861	0.003	0.897	0.003	1.13 × 10^−4^
8	350	91.9	0.861	0.003	0.897	0.003	1.13 × 10^−4^
**9**	**400**	**99.7**	**0.913**	**0.002**	**0.895**	**0.003**	**6.45 × 10^−5^**

**Table 8 sensors-19-01407-t008:** The average of the obtained results in each experiment.

Measurement Period	Accuracy (%)	Validation		
		Linear Correlation	MAE	MSE
May	63.1	0.78	7.44 × 10^−3^	1.51 × 10^−4^
November	72.2	0.848	1.30 × 10^−2^	1.45 × 10^−3^
May 6 to 13	94.7	0.967	1.33 × 10^−3^	7.52 × 10^−6^
November 6 to 13	89.9	0.939	5.00 × 10^−3^	1.17 × 10^−4^
May 12	95.7	0.940	3.12 × 10^−2^	6.14 × 10^−3^
**May 15**	**99.5**	**0.995**	**1.78 × 10^−3^**	**2.44 × 10^−5^**
November 15	92.0	0.912	3.22 × 10^−3^	9.67 × 10^−5^

**Table 9 sensors-19-01407-t009:** The average of the obtained results in the six performed experiments.

Order of Measurement (Model Number)	Number of Neurons (-)	Accuracy (%)	Validation		
			Linear Correlation	MAE	MSE
1	10	64.0	0.774	1.94 × 10^−2^	1.83 × 10^−3^
2	50	87.8	0.899	8.88 × 10^−3^	6.38 × 10^−4^
3	100	82.4	0.925	7.29 × 10^−3^	6.75 × 10^−4^
4	150	89.8	0.927	9.29 × 10^−3^	1.18 × 10^−3^
5	200	91.2	0.935	8.15 × 10^−3^	1.02 × 10^−3^
6	250	91.1	0.931	9.14 × 10^−3^	1.45 × 10^−3^
7	300	91.2	0.935	7.71 × 10^−3^	1.15 × 10^−3^
8	350	90.2	0.936	7.71 × 10^−3^	1.15 × 10^−3^
**9**	**400**	**92.1**	**0.936**	**7.85 × 10^−3^**	**1.17 × 10^−3^**

**Table 10 sensors-19-01407-t010:** The summarization of the evaluation parameters of the original CO_2_ prediction.

Date of Prediction	MSE [-]	Corr [%]	ED [-]
6–13 May 2018	7.577 × 10^−6^	97.0	0.266
15 May 2018	2.436 × 10^−5^	99.5	0.177
1–29 May 2018	1.505 × 10^−4^	77.9	2.439
6–13 November 2018	1.172 × 10^−4^	94.2	1.017
15 November 2018	9.761 × 10^−5^	93.3	0.366
4 November–3 December 2018	4.065 × 10^−4^	79.8	3.429

**Table 11 sensors-19-01407-t011:** The summarization of the evaluation parameters of the wavelet smoothing CO_2_ prediction.

Date of Prediction	MSE [-]	Corr [%]	ED [-]
6–13 May 2018	6.442 × 10^−6^	98.1	0.238
15 May 2018	1.473 × 10^−6^	99.6	0.157
1–29 May 2018	1.360 × 10^−4^	80.6	2.292
6–13 November 2018	1.047 × 10^−4^	95.6	0.925
15 November 2018	7.061 × 10^−5^	95.7	0.315
4 November–3 December 2018	3.781 × 10^−4^	81.3	3.281

**Table 12 sensors-19-01407-t012:** The difference parameters of CO_2_ prediction.

Date of Prediction	Diff MSE [-]	Diff Corr [%]	Diff ED [-]
6–13 May 2018	1.134 × 10^−6^	0.0049	0.0277
15 May 2018	4.886 × 10^−6^	8.524 × 10^−4^	0.0196
1–29 May 2018	1.443 × 10^−5^	0.027	0.147
6–13 November 2018	1.258 × 10^−5^	0.0079	0.092
15 November 2018	2.699 × 10^−5^	0.022	0.051
4 November–3 December 2018	2.842 × 10^−5^	0.015	0.147

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
