# Peer review of "Using the IBM SPSS SW Tool with Wavelet Transformation for CO2 Prediction within IoT in Smart Home Care"

_sensors, 2019, doi:10.3390/s19061407_

Reviewer 1 Report

The authors have tried to provide a methodology that specifies a procedure for processing data measured by sensors in an SHC environment. This indirectly monitors the presence of people in an SHC –rooms (area) through KNX and BACnet technologies commonly applied in building automation.

However, there are certain intricacies in the implementation and needs justification and updating the contents accordingly:

Comprehensive technologies details (existing) with various features in the AAL environments were presented, but not related to the work performed for comparison. Intro to IBM SPSS features can be reduced.

The visualization, monitoring, and processing of the measured values of non-electric variables,  such as measurement of temperature, humidity and CO2 for monitoring the quality of the indoor environment of the selected room were presented, but what is the rationale for the selection of these parameters?

Page 2: lines 48-55::Beaudin,Booysen and Fleck=> Beaudin et al.,

References (53,54,55…)  are not in proper format

The methods related to NEURAL NETWORKS MODEL Ref:56, ACCURACY (Eq.2) Ref:55 (Eq.3) the CO2 concentration prediction optimization based  on the Wavelet transformation additive noise canceling are existing what is the novelty in the present research study?

level of decomposition and the Wavelet settings:  threshold selection rule - Stein's Unbiased Risk and soft thresholding for selection of the detailed  coefficients.??

Therefore, an external program is required in order to perform the filtration. What is meant by an external program??

The CO2 prediction contains lots of significant occurrences represented by the glitches and spikes, significantly deteriorating a smoothness of the analyzed signal. Is the wavelet implementation apt??.

Overall, this is an offline data processing rather than real-time analysis?. Justify the performed process.

Author Response

Response to Reviewer 1 Comments

Topic:

Using the IBM SPSS SW Tool with Wavelet Transformation for CO2 prediction within IoT in Smart Home Care

Authors

Jan Vanus1,*, Jan Kubicek1, Ojan Gorjani1, Jiri Koziorek1

1   Department of Cybernetics and Biomedical Engineering, Faculty of Electrical Engineering and Computer Science, VSB–Technical University of Ostrava, Ostrava, Czech Republic; jan.vanus@vsb.cz; jan.kubicek@vsb.cz; ojan.majidzadeh.gorjani@vsb.cz, jiri.koziorek@vsb.cz 

*   Correspondence: jan.vanus@vsb.cz; Tel.: +420-59-732-5856

5/03/2018

First of all, we would like to thank the reviewer for the constructive remarks on the first version of this paper. We think that the new version of our manuscript includes all the suggested changes. We believe that the quality of the present version has improved greatly; it adequately addresses all the comments and provides clear explanations.

Below we provide a brief account of how the issues pointed out by the reviewer have been addressed in the paper.

The authors have tried to provide a methodology that specifies a procedure for processing data measured by sensors in an SHC environment. This indirectly monitors the presence of people in an SHC –rooms (area) through KNX and BACnet technologies commonly applied in building automation. However, there are certain intricacies in the implementation and needs justification and updating the contents accordingly:

Comprehensive technologies details (existing) with various features in the AAL environments were presented, but not related to the work performed for comparison.

Response 1: Sentence (line 37-41):

The use of technology to improve the people quality of life is becoming a common trait of modern society. When the technology is oriented to improve the QoL at Smart Home, it is referred to as Ambient Assisted Living (AAL). AAL technologies are typically classified according to the specific needs for which they are developed, in particular: physical and physiological needs, safety, security and comfort needs, and autonomy needs [3].

was exchanged with the sentence:

Privacy, reliability and false alarms are the main challenges to be considered for the development of efficient systems to detect and classify the Activities of Daily Living (ADL) and Falls [3].

Intro to IBM SPSS features can be reduced.

Response 2: Intro to IBM SPSS was reduced from original (91-138) to final version (88-89) [23 - 35].

The visualization, monitoring, and processing of the measured values of non-electric variables,  such as measurement of temperature, humidity and CO2 for monitoring the quality of the indoor environment of the selected room were presented, but what is the rationale for the selection of these parameters?

Response 3: The rationale for the selection of these parameters is, that these sensors are common-used sensors for operational and technical HVAC (Heating, Ventilation, and Air conditioning) functions within Building automation indoor environment quality control. The CO2 sensor is monitoring a room person´s occupancy in the monitored area of Smart Home, arrival and departure.

Page 2: lines 48-55::Beaudin,Booysen and Fleck=> Beaudin et al.,

Response 4: Page 2, lines 48 – 55  Beaudin et al., Booysen et al., Basu et al.,  Fleck et al., AlFaris et al., Alirezaie et al., Bassoli et al., Catherwood et al.

References (53,54,55…)  are not in proper format

Response 5: References (53,54,55…)  were repaired and now are in proper format

 53. IBM SPSS Modeler 16 Algorithms Guide. Available online: ftp://public.dhe.ibm.com/software/analytics/spss/documentation/modeler/16.0/en/AlgorithmsGuide.pdf (accessed on 25/11/2018).

54. IBM SPSS Modeler 17 Algorithms Guide. Available online: ftp://public.dhe.ibm.com/software/analytics/spss/documentation/modeler/17.0/en/AlgorithmsGuide.pdf (accessed on 25/11/2018).

55. IBM SPSS Modeler 18 Algorithms Guide. Available online: ftp://public.dhe.ibm.com/software/analytics/spss/documentation/modeler/18.0/en/AlgorithmsGuide.pdf (accessed on 25/11/2018).

56. PI System™ From data to knowledge to transformation. Available online: https://www.osisoft.com/pi-system/#tab1 (accessed on 10/01/2019).

The methods related to NEURAL NETWORKS MODEL Ref: 56, ACCURACY (Eq.2) Ref:55 (Eq.3) the CO2 concentration prediction optimization based on the Wavelet transformation additive noise cancelling are existing what is the novelty in the present research study?

Response 6:

In this paper, we present a robust hybrid system for the CO2 concentration prediction. The system is based on the neural network, which performs prediction of the CO2 concentration. Based on the experimental results, we have found out that the predicted signal is accompanied by rapid oscillations (glitches) significantly deteriorate the predicted signals. Thus, this phenomenon has a negative impact on CO2 prediction accuracy. Therefore, our hybrid system is completed with the wavelet filtration, which is able to suppress these glitches and make the predicted signal much smoother, when comparing with the original prediction. Prediction accuracy increase is supported by the objective evaluation based on the evaluation parameters (MSE, Euclidean distance and correlation coefficient). Such complex method utilizing a hybrid combination of the neural network and the wavelet filtration makes a novel and robust system for the CO2 prediction for the smart home monitoring.

Level of decomposition and the Wavelet settings:  threshold selection rule - Stein's Unbiased Risk and soft thresholding for selection of the detailed coefficients? 

Response 7:

Wavelet transforms the CO2 signal on the sequence of the wavelet coefficients. This sequence represents the CO2 signal in the Wavelet domain, which contains oscillations which should be suppressed for better prediction accuracy. Appropriate thresholding and consequent elimination of the Wavelet coefficients representing the glitches aims to the CO2 prediction smoothing and improving prediction.  We experimentally selected mother Wavelet Db6, with 6-level of decomposition for the signal transformation. The Wavelet performs filtration based on the Wavelet coefficients thresholding. For the thresholding, we use the selection rule - Stein's Unbiased Risk for determining the threshold and soft thresholding of the wavelet coefficients for the thresholding.  Alternatively, against Stein's Unbiased Risk and soft thresholding, we have tested the universal threshold and hard thresholding. Based on the objective evaluation (correlation index and MSE) for the CO2 prediction, we use the optimal settings for CO2 signal detection (Wavelet Db6, with 6-level of decomposition, Stein's Unbiased Risk soft thresholding), as we state in the paper.

Therefore, an external program is required in order to perform the filtration. What is meant by an external program?? 

Response 8: External Software tool may be the more appropriate term, this sentence had been removed.

The CO2 prediction contains lots of significant occurrences represented by the glitches and spikes, significantly deteriorating a smoothness of the analyzed signal. Is the wavelet implementation apt??.

Response 9:

Wavelet transformation represents a powerful tool for the signal decomposition and filtration of the predicted CO2 signal due to the extensive setting (selection of wavelet type, decomposition level, thresholding rule and type of thresholding). Based on the experimental results reported in the manuscript, the Wavelet transformation allows for the prediction optimization. These outputs are supported by the objective evaluation based on the MSE, Euclidean distance and correlation index. Furthermore, we can compare the Wavelet transformation with the adaptive filtering [1] which was used for the CO2 signal filtration in the previous study. In comparison with the adaptive filtration, Wavelet transformation does not require a reference signal for filtering. This fact makes from Wavelet method more robust method for the filtration.

[1] Vanus, J., Martinek, R., Nedoma, J., Fajkus, M., Cvejn, D., Valicek, P., Novak, T. Utilization of the LMS Algorithm to Filter the Predicted Course by Means of Neural Networks for Monitoring the Occupancy of Rooms in an Intelligent Administrative Building (2018) IFAC-PapersOnLine, 51 (6), pp. 378-383.

Overall, this is an offline data processing rather than real-time analysis? Justify the performed process.

Response 10: As it was described in section 4.2.5. At this moment online data processing is possible, however, we are working on a few minor database connectivity issues. Once these issues are resolved this implementation can perform near-real-time online data processing. 

On the behalf of all the authors, I would like to once again, express our gratitude for the reviewer’s support and advice.

Best regards,

Jan Vanus

Reviewer 2 Report

This manuscript examines the possibilities of increasing the accuracy of CO2 predictions in Smart Home Care using IBM SPSS software tools in the IoT to determine the occupancy times of a monitored SHC room, based on the measurements from the indoor and outdoor temperatures and relative humidity. This is a fine contribution to the literature on this subject and is definitely worthy of publication. I thoroughly enjoyed reading this manuscript. I have only minor corrections/comments that I think will clarify the text:

(1)     Line 261-264, page 7. The sensing accuracies of the two selected indoor and outdoor temperature sensors are quite different, 1K and 0.1K, respectively. The reviewer is wondering why the authors choose this sensor combination, and would this make any differences for the analysis and prediction carried out later on? And for the relative humidity sensor, the typical sensing accuracy for commercial sensor should be 1-2%. Again, would 1% accuracy relative humidity sensor help to improve the prediction? What are the sampling frequencies for all these three sensors? Are they sampling at the same frequency and are they synchronised?

(2)     Equation 1, page 8, may miss right bracket?

(3)     Line 293, page 8. ‘high frequency part of the signal’, what is the frequency of this high frequency component? Does this related to the sampling frequency mentioned in (1);

(4)     Fig. 5, page 11. What is E (Ix) in the figure? Figures 5-8 presents the history data of CO2, indoor temperature, Blinds and Slats. How about the history data for relative humidity and outdoor temperature, as discussed before?

(5)     Line 394, page 11. ‘VI. The 393 window was opened partially at 11:36:07, VII. The window was closed at 11:36:07.’ The timing for VII may not be correct.

(6)     Line 513, page 16. ‘the highest linear correlation (0.998)’ may not be correct, according to table 7.

(7)     Figure 12, page 19. The reviewer is curious about the selection of 15th May. Firstly, 15th May2018 is a Tuesday, why the accumulation of CO2 started in the late afternoon, rather different from the ones presented in Figs. 5-8. Secondly, the two history data (Fig. 12 and Fig. 5) have very different pattern, and obviously, the one shown in Fig. 5 should be much more common. Does this also imply that the proposed prediction methodology works better for the ones similar to Fig. 12? If not, the authors should perform further comparison for other days to demonstration the capability of the proposed method.

(8)     What is the unit of the vertical axis in Fig. 12 and Fig. 13? May be better to be in ppm, as shown in Figs. 5-8.

Author Response

Response to Reviewer 2 Comments

Topic:

Using the IBM SPSS SW Tool with Wavelet Transformation for CO2 prediction within IoT in Smart Home Care

Authors

Jan Vanus1,*, Jan Kubicek1, Ojan Gorjani1, Jiri Koziorek1

1   Department of Cybernetics and Biomedical Engineering, Faculty of Electrical Engineering and Computer Science, VSB–Technical University of Ostrava, Ostrava, Czech Republic; jan.vanus@vsb.cz; jan.kubicek@vsb.cz; ojan.majidzadeh.gorjani@vsb.cz, jiri.koziorek@vsb.cz 

*   Correspondence: jan.vanus@vsb.cz; Tel.: +420-59-732-5856

5/03/2018

First of all, we would like to thank the reviewer for the constructive remarks on the first version of this paper. We think that the new version of our manuscript includes all the suggested changes. We believe that the quality of the present version has improved greatly; it adequately addresses all the comments and provides clear explanations.

Below we provide a brief account of how the issues pointed out by the reviewer have been addressed in the paper.

This manuscript examines the possibilities of increasing the accuracy of CO2 predictions in Smart Home Care using IBM SPSS software tools in the IoT to determine the occupancy times of a monitored SHC room, based on the measurements from the indoor and outdoor temperatures and relative humidity. This is a fine contribution to the literature on this subject and is definitely worthy of publication. I thoroughly enjoyed reading this manuscript. I have only minor corrections/comments that I think will clarify the text:

(1)        Line 261-264, page 7. The sensing accuracies of the two selected indoor and outdoor temperature sensors are quite different, 1K and 0.1K, respectively. The reviewer is wondering why the authors choose this sensor combination, and would this make any differences for the analysis and prediction carried out later on? And for the relative humidity sensor, the typical sensing accuracy for commercial sensor should be 1-2%. Again, would 1% accuracy relative humidity sensor help to improve the prediction? What are the sampling frequencies for all these three sensors? Are they sampling at the same frequency and are they synchronised?

Response 1: Outdoor sensor To (°C), (AP 257/22) is KNX sensor implemented in Weather station for operational and technical functions control in Smart Home. For HVAC control in SH is used BACnet technology - (QPA 2062 sensor). There is interoperability between KNX and BACnet technology through KNX/BACnet gateway. Visualization is performed in DESIGO INSIGHT SW tool. For data, saving is in DESIGO INSIGHT irregular interval data storage. In SW Tool PI System, which is connected with DESIGO INSIGHT through PI OPC BACnet interface, is possible settings of the desired interval for data processing with soft-computing methods. In the next experiments, we will use sensors with higher accuracy with possible settings of the desired interval in low cost solving. Interval storing data from all three sensors is 1 minute. Data of all three sensors are synchronized for data processing.

(2)        Equation 1, page 8, may miss right bracket?

Response 2: Equation 1 was repaired with right bracket.

(3)        Line 293, page 8. ‘high frequency part of the signal’, what is the frequency of this high frequency component? Does this related to the sampling frequency mentioned in (1); 

Response 3: In this sentence (line 293), we wanted to point out that the predicted CO2 signal contains rapid oscillations, so called glitches, which do not have the origin in the original CO2 signal. Due to presence of these occurrences, we employed the Wavelet filtration to obtain a smooth trend of the CO2 signal, while these oscillations are removed. We maybe used the inappropriate term ‘’high frequency component’’ for these oscillations. We wanted to express that these signal segments have oscillating character. In our system, we use the sampling frequency 1Hz/minute, which is enough for the Smart home operational and technical functions measurement and control. We reformulate the sentence from the line 293. Nevertheless, we also devoted to the time-frequency analysis of the CO2 signal to mapping the signal stationarity within the time and discover frequency spectrum of the mentioned oscillating glitches. We subtracted the original CO2 signal from the filtered signal to identify parts of the activity corresponds with these oscillations which are removed by the Wavelet. Consequently, we applied the STFT (Short-Time Fourier Transform,) to locate the time-frequency PSD (Power Spectral Density) spectrum of these oscillations. Based on the results published in the manuscript, we estimate the average frequency range: 0.8;1 Hz.

Original sentence (line 293): Based on the experimental results, the predicted CO2 trend contains glitches representing high-frequency part of the signal.

Modified sentence: (line 244): Based on the experimental results, the predicted CO2 trend contains glitches representing fast change part of the signal.

(4)     Fig. 5, page 11. What is E (Ix) in the figure? Figures 5-8 presents the history data of CO2, indoor temperature, Blinds and Slats. How about the history data for relative humidity and outdoor temperature, as discussed before?

Response 4: E (lx) in the figure 5. is Illuminance waveform

Figure. 5 The reference (measured) CO2 concentration course and Illuminance E (lx) course (12.11.2018) in room R203 in SHC for ADL monitoring.

History data for relative humidity and outdoor temperature were added to the Figure 6.

(5)     Line 394, page 11. ‘VI. The 393 window was opened partially at 11:36:07, VII. The window was closed at 11:36:07.’ The timing for VII may not be correct

Response 5: The time for VII point (Figure 6) was changed - 14:00:35.

(6)     Line 513, page 16. ‘the highest linear correlation (0.998)’ may not be correct, according to table 7.

Response 6:  It had been corrected (“relevantly high linear correlation value (0.895)”)

(7)     Figure 12, page 19. The reviewer is curious about the selection of 15th May. Firstly, 15th May2018 is a Tuesday, why the accumulation of CO2 started in the late afternoon, rather different from the ones presented in Figs. 5-8. Secondly, the two history data (Fig. 12 and Fig. 5) have very different pattern, and obviously, the one shown in Fig. 5 should be much more common. Does this also imply that the proposed prediction methodology works better for the ones similar to Fig. 12? If not, the authors should perform further comparison for other days to demonstration the capability of the proposed method.

Response 7: The intervals of the experiment were chosen at random. The pattern of the signal does not have a major impact on the results. The experimenter had been repeated for 12 of May with the following results:

Table 6. Trained models – Training period: 12th of May 2018

Order of measurement

(Model Number)

Number of neurons(-)

Accuracy

(%)

testing

validation

Linear correlation

MAE

Linear correlation

MAE

MSE

1

10

88.9

0.927

0.064

0.922

0.067

9.81*10-3

2

50

97.8

0.972

0.026

0.977

0.024

2.63*10-3

3

100

98.0

0.971

0.022

0.967

0.021

3.11*10-3

4

150

96.7

0.940

0.037

0.934

0.038

6.54*10-3

5

200

98.1

0.937

0.035

0.943

0.032

5.43*10-3

6

250

95.0

0.921

0.035

0.901

0.040

8.48*10-3

7

300

95.7

0.941

0.031

0.932

0.030

6.42*10-3

8

350

95.7

0.941

0.031

0.935

0.030

6.42*10-3

9

400

95.7

0.941

0.031

0.935

0.030

6.42*10-3

The model number 3 trained and validated using data from 12th of May 2018.

Further comparisons of implementation of designed method are in:

J. Vanus et al., "The design of an indirect method for the human presence monitoring in the intelligent building," Human-centric Computing and Information Sciences, Article vol. 8, no. 1, 2018, Art no. 28, doi: 10.1186/s13673-018-0151-8.

J. Vanus, R. Martinek, P. Bilik, J. Zidek, P. Dohnalek, and P. Gajdos, "New method for accurate prediction of CO2 in the Smart Home," in Conference Record - IEEE Instrumentation and Measurement Technology Conference, 2016, vol. 2016-July, doi: 10.1109/I2MTC.2016.7520562.

J. Vanus, R. Martinek, J. Kubicek, M. Penhaker, J. Nedoma, and M. Fajkus, "Using the PI processbook software tool to monitor room occupancy in smart home care," in 2018 IEEE 20th International Conference on e-Health Networking, Applications and Services, Healthcom 2018, 2018, doi: 10.1109/HealthCom.2018.8531108.

J. Vanus et al., "Utilization of the LMS Algorithm to Filter the Predicted Course by Means of Neural Networks for Monitoring the Occupancy of Rooms in an Intelligent Administrative Building," IFAC-PapersOnLine, Article vol. 51, no. 6, pp. 378-383, 2018, doi: 10.1016/j.ifacol.2018.07.183.

J. Vanus, R. Martinek, J. Nedoma, M. Fajkus, P. Valicek, and T. Novak, "Utilization of Interoperability between the BACnet and KNX Technologies for Monitoring of Operational-Technical Functions in Intelligent Buildings by Means of the PI System SW Tool," IFAC-PapersOnLine, Article vol. 51, no. 6, pp. 372-377, 2018, doi: 10.1016/j.ifacol.2018.07.182.

J. Vanus et al., "Monitoring of the daily living activities in smart home care," Human-centric Computing and Information Sciences, Article vol. 7, no. 1, 2017, Art no. 30, doi: 10.1186/s13673-017-0113-6.

(8) What is the unit of the vertical axis in Fig. 12 and Fig. 13? May be better to be in ppm, as shown in Figs. 5-8.

Response 8: As it was explained in section 4.2.1. Pre-processing, prior to the analysis, the data were normalized (using min-max method), therefore the CO2 values do not have any unit. The normalization had been reversed and CO2 values had been converted back to ppm (Fig.12 – Fig.14).

On the behalf of all the authors, I would like to once again, express our gratitude for the reviewer’s support and advice.

Best regards,

Jan Vanus

Reviewer 3 Report

Related work describes many in a global smart home domain, more similar or CO2-related references are expected.

This is good work.

Table 1 is not informative so that it can be described in a sentence.

Cyan font color in Table 2-9 can be replaced in bold font style.

Author Response

Response to Reviewer 3 Comments

Topic:

Using the IBM SPSS SW Tool with Wavelet Transformation for CO2 prediction within IoT in Smart Home Care

Authors

Jan Vanus1,*, Jan Kubicek1, Ojan Gorjani1, Jiri Koziorek1

1   Department of Cybernetics and Biomedical Engineering, Faculty of Electrical Engineering and Computer Science, VSB–Technical University of Ostrava, Ostrava, Czech Republic; jan.vanus@vsb.cz; jan.kubicek@vsb.cz; ojan.majidzadeh.gorjani@vsb.cz, jiri.koziorek@vsb.cz 

*   Correspondence: jan.vanus@vsb.cz; Tel.: +420-59-732-5856

5/03/2018

First of all, we would like to thank the reviewer for the constructive remarks on the first version of this paper. We think that the new version of our manuscript includes all the suggested changes. We believe that the quality of the present version has improved greatly; it adequately addresses all the comments and provides clear explanations.

Below we provide a brief account of how the issues pointed out by the reviewer have been addressed in the paper.

Comments and Suggestions for Authors

Related work describes many in a global smart home domain, more similar or CO2-related references are expected.

Response 1: Related references are here:

Z. Liu, K. Cheng, H. Li, G. Cao, D. Wu, and Y. Shi, "Exploring the potential relationship between indoor air quality and the concentration of airborne culturable fungi: a combined experimental and neural network modeling study," Environmental Science and Pollution Research, Article vol. 25, no. 4, pp. 3510-3517, 2018, doi: 10.1007/s11356-017-0708-5.

M. Sterman and M. Baglione, "Simulating the use of CO2 concentration inputs for controlling temperature in a hydronic radiant system," in ASME International Mechanical Engineering Congress and Exposition, Proceedings (IMECE), 2017, vol. 6, doi: 10.1115/IMECE2017-71095.

M. S. Zuraimi, A. Pantazaras, K. A. Chaturvedi, J. J. Yang, K. W. Tham, and S. E. Lee, "Predicting occupancy counts using physical and statistical Co2-based modeling methodologies," Building and Environment, Article vol. 123, pp. 517-528, 2017, doi: 10.1016/j.buildenv.2017.07.027.

J. Vanus et al., "The design of an indirect method for the human presence monitoring in the intelligent building," Human-centric Computing and Information Sciences, Article vol. 8, no. 1, 2018, Art no. 28, doi: 10.1186/s13673-018-0151-8.

J. Vanus, R. Martinek, P. Bilik, J. Zidek, P. Dohnalek, and P. Gajdos, "New method for accurate prediction of CO2 in the Smart Home," in Conference Record - IEEE Instrumentation and Measurement Technology Conference, 2016, vol. 2016-July, doi: 10.1109/I2MTC.2016.7520562.

J. Vanus, R. Martinek, J. Kubicek, M. Penhaker, J. Nedoma, and M. Fajkus, "Using the PI processbook software tool to monitor room occupancy in smart home care," in 2018 IEEE 20th International Conference on e-Health Networking, Applications and Services, Healthcom 2018, 2018, doi: 10.1109/HealthCom.2018.8531108.

J. Vanus et al., "Utilization of the LMS Algorithm to Filter the Predicted Course by Means of Neural Networks for Monitoring the Occupancy of Rooms in an Intelligent Administrative Building," IFAC-PapersOnLine, Article vol. 51, no. 6, pp. 378-383, 2018, doi: 10.1016/j.ifacol.2018.07.183.

J. Vanus, R. Martinek, J. Nedoma, M. Fajkus, P. Valicek, and T. Novak, "Utilization of Interoperability between the BACnet and KNX Technologies for Monitoring of Operational-Technical Functions in Intelligent Buildings by Means of the PI System SW Tool," IFAC-PapersOnLine, Article vol. 51, no. 6, pp. 372-377, 2018, doi: 10.1016/j.ifacol.2018.07.182.

J. Vanus et al., "Monitoring of the daily living activities in smart home care," Human-centric Computing and Information Sciences, Article vol. 7, no. 1, 2017, Art no. 30, doi: 10.1186/s13673-017-0113-6.

This is good work.

Table 1 is not informative so that it can be described in a sentence.

Response 2: The table1 was removed and described in the text.

Cyan font color in Table 2-9 can be replaced in bold font style.

Response 3: Cyan color in tables replaced with bold font.

On the behalf of all the authors, I would like to once again, express our gratitude for the reviewer’s support and advice.

Best regards,

Jan Vanus

Round  2

Reviewer 1 Report

Authors have incorporated my suggestions as prescribed in the previous review. Response 6,7,8 are acceptable but Response 9 requires valid justification. 

Author Response

2 Response to Reviewer 1 Comments

Topic:

Using the IBM SPSS SW Tool with Wavelet Transformation for CO2 prediction within IoT in Smart Home Care

Authors

Jan Vanus1,*, Jan Kubicek1, Ojan Gorjani1, Jiri Koziorek1

1   Department of Cybernetics and Biomedical Engineering, Faculty of Electrical Engineering and Computer Science, VSB–Technical University of Ostrava, Ostrava, Czech Republic; jan.vanus@vsb.cz; jan.kubicek@vsb.cz; ojan.majidzadeh.gorjani@vsb.cz, jiri.koziorek@vsb.cz 

*   Correspondence: jan.vanus@vsb.cz; Tel.: +420-59-732-5856

7/03/2018

First of all, we would like to thank the reviewer for the constructive remarks on the first version of this paper. We think that the new version of our manuscript includes all the suggested changes. We believe that the quality of the present version has improved greatly; it adequately addresses all the comments and provides clear explanations.

Below we provide a brief account of how the issues pointed out by the reviewer have been addressed in the paper.

Comments and Suggestions for Authors:

Authors have incorporated my suggestions as prescribed in the previous review. Response 6,7,8 are acceptable but Response 9 requires valid justification.

9. The CO2 prediction contains lots of significant occurrences represented by the glitches and spikes, significantly deteriorating a smoothness of the analyzed signal. Is the wavelet implementation apt??.

Response 9:

In the article we added the following paragraph:

Alternatively, we mention the comparison [50] of the CO2 filtration based on the LMS algorithm. In this study, the authors employed adaptive filtration. The main limitation of this method is a necessity of the reference signal and a slow adaptation of the filtration procedure as well depending on the accuracy of the step size parameter μ calculation and inaccurate determination of the arrival and departure time of the person from the monitored area. Furthermore, the Wavelet filtration presented in this study achieves better results in a context of the objective comparison against the LMS filtration. Wavelet filtration has a much stronger potential for the CO2 filtration due to a possibility of application of variety wavelets allowing for extraction of specific morphological signal features in various decomposition levels and thus, better optimization of the CO2 prediction. These facts predetermine wavelets to be a robust system for the CO2 prediction enhancement.

Wavelet transformation represents a powerful tool for the signal decomposition and filtration of the predicted CO2 signal due to the extensive setting (selection of wavelet type, decomposition level, thresholding rule and type of thresholding). Based on the experimental results reported in the manuscript, the Wavelet transformation allows for the prediction optimization. These outputs are supported by the objective evaluation based on the MSE, Euclidean distance and correlation index. Furthermore, we can compare the Wavelet transformation with the adaptive filtering [1] which was used for the CO2 signal filtration in the previous study. In comparison with the adaptive filtration, Wavelet transformation does not require a reference signal for filtering. This fact makes from Wavelet method more robust method for the filtration [2].

For comparison, we made experiments with LMS adaptive filter algorithm in the article [1] (https://hcis-journal.springeropen.com/articles/10.1186/s13673-018-0151-8):

The short-term experiment-7 days. Training the ANN with the BRM was implemented on the data (from February 8 to February 14, 2015). 1 week (08/02/2015–14/02/2015).

Fig. 26 The reference [measured course of CO2 concentration (from February 8 to February 14, 2015)] and the predicted course of CO2 concentration. The ANN with the BRM learned (from June 8 to June 14, 2015), the predicted ANN with the data from February 8 to February 14, 2015—the number of neurons—400. 1. Arrival (8:50:00), 2. departure (10:30:00), TPP ∆t1 = 1:40:00; 3. arrival (10:50:00), 4. departure (11:10:00), TPP ∆t2 = 0:20:00; 5. arrival (11:20:00) and 6. departure (11:30:00), TPP ∆t3 = 0:20:00

Figure 27 illustrates the reference measured course of CO2 concentration from February 8 to February 14, 2015. Figure 27 further illustrates the predicted filtered course of CO2concentration using the LMS algorithm [the ANN with the BRM learned on the data (from June 8 to June 14, 2015) and predicted within the cross-validation (Step 6c) with the data from February 8 to February 14, 2015, using the ANN with the BRM (the number of neurons—400)]. Based on relations (6) to (11), the step size parameter μ values for setting the structure of the LMS algorithm adaptive filter (M = 44, μ = 5.63·10−2) for noise elimination in the calculated predicted course of CO2 (Fig. 26).

Fig. 27

The reference [measured course of CO2 concentration (from February 8 to February 14, 2015)] and predicted filtered course of CO2 concentration using the LMS algorithm (the ANN with the BRM learned on the data (from June 8 to June 14, 2015) and predicted within the cross-validation with the data from February 8 to February 14, 2015, using the ANN with the BRM (the number of neurons—400). 1. Arrival (8:50:00), 2. departure (10:30:00), TPP ∆t1 = 1:40:00; 3. arrival (10:50:00), 4. departure (11:10:00), TPP ∆t2 = 0:20:00; 5. arrival (11:20:00) and 6. departure (11:30:00), TPP ∆t3 = 0:20:00

Discussion

Experiment

The achieved results confirmed the improvement in the predicted course of CO2 by means of ANN BRM using the LMS algorithm. Distance D calculated by means of the DTW criterion (13) and (14D = 0.256 was higher for the predicted course of CO2(Fig. 26) not using the LMS algorithm. Similarly, the correlation coefficient R = 0.89 was lower. When the LMS algorithm was used (Fig. 27), the distance between the compared courses was smaller, D = 0.107. Similarly, the calculated correlation coefficient R = 0.93 was higher. The advantage of the proposed method consists in the use of common operating sensors to obtain information on the state of the operational-technical functions in the Intellient Administrative Building (IAB) for the purpose of their optimum control on the basis of predictable needs of the persons using the IAB areas. The disadvantage of the proposed method with the LMS algorithm is the slow adaptation of the LMS algorithm to real conditions depending on the precision of calculation of the step size parameter μ and inaccurate determination of a person’s arrival and departure to and from the monitored area (Fig. 27), (between points 2 and 4). This is, however, a small decline in the CO2concentration (ppm) depending on the quick departure and arrival of the person. The proposed method is not able to record the quick change due to the small or big changes in the CO2 concentration. On the basis of the above-indicated results, however, one can state that the LMS algorithm has brought improved precision of the proposed method.

Due to the complexity of the ANN BRM mathematical model, the learning process is a non-trivial optimization problem. Iterative algorithms often encounter the local minima in which the learning process is terminated. The ANN instability is a consequence of such behaviour. Especially the fact that two learning processes run on the same data (but from different starting points) end in two different local solutions is often criticized. Partial elimination of this disadvantage is ensured by using a corresponding adaptive filter. The advantage of the method designed is the possible use of common operational sensors for the detection of information on the state of operational and technical functions in the IAB. The disadvantage of the method designed with the LMS algorithm is the slow adaptation of the LMS algorithm to real conditions, depending on the accuracy of the step size parameter μ calculation and inaccurate determination of the arrival and departure time of the person from the monitored area.

Conclusion

The article has verified using the ANN with BRM to predict CO2 for the purposes of monitoring the presence of people in the selected IAB office. In addition, the article verified the improvement of the reliability and accuracy of the proposed indirect method using the LMS adaptive filter for filtering the predicted CO2 concentration course for seasons in a time interval between winter and spring (from February 1 to February 28, 2015), for a time interval between spring and summer (from June 1 to June 28, 2015) and for short experiment in winter (from February 8 to February 14, 2015).

In the article, we made moderate English changes (we fixed the errors found in English).

References

[1] Vanus, J., J. Machac, R. Martinek, P. Bilik, J. Zidek, J. Nedoma & M. Fajkus (2018a) The design of an indirect method for the human presence monitoring in the intelligent building. Human-centric Computing and Information Sciences, 8.

[2] Vanus, J., Martinek, R., Nedoma, J., Fajkus, M., Cvejn, D., Valicek, P., Novak, T. Utilization of the LMS Algorithm to Filter the Predicted Course by Means of Neural Networks for Monitoring the Occupancy of Rooms in an Intelligent Administrative Building (2018) IFAC-PapersOnLine, 51 (6), pp. 378-383.

On the behalf of all the authors, I would like to once again, express our gratitude for the reviewer’s support and advice.

Best regards,

Jan Vanus

This manuscript is a resubmission of an earlier submission. The following is a list of the peer review reports and author responses from that submission.